# Understory plants evade shading in a temperate deciduous forest amid climate variability by shifting phenology in synchrony with canopy trees

Carol K. Augspurger[1]☝, Carl F. Salk[2,3]☝ *

1 Department of Plant Biology, University of Illinois, Urbana, IL, United States of America, 2 Southern Swedish Forest Research Centre, Swedish University of Agricultural Sciences, Alnarp, Sweden, 3 Institute for Globally-Distributed Open Research and Education, Gothenburg, Sweden

☝ These authors contributed equally to this work.
* carl.salk@slu.se

**Data Availability Statement:** All data used in this analysis can be found in Dryad at doi:10.5061/dryad.pg4f4qrww.

## Abstract

Global warming is leading understory and canopy plant communities of temperate deciduous forests to grow leaves earlier in spring and drop them later in autumn. If understory species extend their leafy seasons less than canopy trees, they will intercept less light. We look for mismatched phenological shifts between canopy and understory in 28 years (1995–2022) of weekly data from Trelease Woods, Urbana, IL, USA. The observations cover 31 herb species of contrasting seasonality (for 1995–2017), three sapling species, and the 15 most dominant canopy tree species for all years, combined with solar radiation, temperature and canopy light transmittance data. We estimate how understory phenology, cold temperatures, canopy phenology, and solar radiation have individually limited understory plants' potential light interception over >2 decades. Understory and canopy phenology were the two factors most limiting to understory light availability, but which was more limiting varied greatly among species and among/within seasonality groups; solar radiation ranked third and cold fourth. Understory and canopy phenology shifts usually occurred in the same direction; either both strata were early or both were late, offsetting each other's effects. The four light-limiting factors combined showed significant temporal trends for six understory species, five toward less light interception. Warmer springs were significantly associated with shifts toward more light interception in three sapling species and 19 herb species. Canopy phenology became more limiting in warmer years for all three saplings species and 31 herb species. However, in aggregate, these variables mostly offset one another; only one sapling and seven herb species showed overall significant (and negative) relationships between light interception and spring temperature. The few understory species mismatched with canopy phenology due to changing climate are likely to intercept less light in future warmer years. The few species with data for carbon assimilation show broadly similar patterns to light interception.

**Funding:** The author(s) received no specific funding for this work.

## Introduction

In temperate deciduous forests, light availability to understory plants varies greatly among seasons as the canopy trees grow leaves in spring, have a full canopy throughout the summer, and drop leaves in autumn [1, 2]. The amount of light available to understory plants depends on seasonal changes in the sun's position in the sky, trunk and branch interference, and the relative leaf phenology of the two forest strata. Shading by branches and trunks in mid-latitude temperate forests can intercept two-thirds of photosynthetically-active solar radiation (PAR) during the winter, and nearly half of PAR in late spring; when the canopy is fully leafed less than 2% of PAR reaches the understory [3]. This seasonally- and phenologically-dependent variation in light availability means that even relatively small global-warming-driven changes in understory and canopy phenology could have big consequences for understory plants' carbon balance [4] and growth and survival [5, 6].

Some understory plants, including seedlings [5] and saplings [7] of canopy trees and herbs [8], have some degree of shade avoidance, i.e., some portion of their yearly leafy period happens before canopy closure in spring and/or after canopy opening in autumn [9–11]. The amount of their total annual irradiance received during this escape to high light ranges widely, from virtually none for summer-active herbs to nearly 100% for winter-active herbs, from 57% to 84% for seedlings of tree species [5] and from 45–98% for five woody understory species [12]. Generally, understory plants intercept more light during pre-canopy closure in spring and less following autumn canopy opening [3, 4], because the sun is lower in the sky during autumn than spring. Shade avoidance brings fitness benefits, including better seedling survival [5]. Experimental studies, either enhancing spring light [13] or increasing shade [14, 15], show that flowering, seed production, growth, and reproduction of understory plants are greater with more spring light.

Climate change is impacting phenological patterns, with warmer, earlier springs leading to earlier phenology of all functional plant groups in temperate deciduous forests [16, 17]. Climate change delays chilling overnight temperatures in autumn, typically resulting in later senescence and leaf drop for all functional groups/species still active [16, 18]. Both of these patterns contribute to a longer growing season.

The phenological mismatch hypothesis, as applied to plant strata, addresses whether the phenology of canopy trees or understory plants is more sensitive to climate warming. The rationale for the different responses to spring warming relates to the two strata requiring different environmental conditions to initiate spring development. Woody plants use air temperature and sometimes photoperiod to cue leafing [19]. Herbs first require warmer soil temperatures to break dormancy of underground structures [11]; later, after emergence, they respond to warming air temperatures. Because air temperatures in spring increase sooner than soil temperatures, tree phenology may advance more than understory herb phenology, resulting in a shorter period of high light for herb species [11]. In contrast, air temperature in spring is higher in the understory than in the canopy [20], so climate warming may affect sapling phenology differently than canopy trees, potentially lengthening or shortening the period of high light for saplings of canopy tree species. Predicting phenological mismatches in autumn is more difficult because, while climate warming is associated with woody plants beginning senescence later [21], no consistent response among late-senescing herb species occurs [8]. Warm temperatures promote senescence of spring herbs [8, 22], but less is known about cues to initiate senescence in understory plants active into summer, autumn, and winter. Saplings presumably experience warmer minimum temperatures than canopy trees in autumn, so may have different changes in senescence patterns than canopy trees.

Although studies are accumulating that document phenological mismatches between trophic levels, e.g., plants and pollinators or herbivores, less is known about contrasting strata within the plant trophic level [23]. Tests of the mismatch hypothesis between plant strata have used common-garden experiments [24], experimental warming chambers [13], herbarium collections [25–27], experimental field studies [5], and direct field observations [16], some by citizen scientists [17] over varying numbers of years. The geographical scale varies from regional [27] to global latitudinal comparisons [26]. The studies vary in number of species and phenological seasonality of herb species. Almost all are based on spring phenology. Studies of the entire growing season are limited to one shrub species [3], seven herb species [4], and seedlings of two tree species [5]. Given the year-to-year and spatial variability of weather, assessing phenological mismatches requires observations of interacting organisms within a single location [28], either in long-term (ideally multi-decade) contemporary data sets or by comparing the same species for multiple years between historic periods [4, 17]. Limited studies of the mismatch hypothesis comparing plant strata for spring are at different geographical scales and are contradictory. Trees are advancing more days than herbs in North America [24], while the opposite pattern is seen in China [16]. Herbs are more thermally sensitive than trees at mid and high latitudes, but not at low latitudes [29]. Herb species are comparable in their sensitivity to spring temperatures across the northern hemisphere, but canopy trees in North America are more sensitive to temperature than those in Asia and Europe [26]. Trees advance leaf out slightly more days/˚C than herbs [27]. Trees in spring have advanced more absolute days historically than herbs [17].

A key underlying assumption of the mismatch hypothesis is that a change in phenological date (or even relative sensitivity to temperature changes) is a good proxy of a change in canopy closure, and, more indirectly, to a change in light availability to the understory. Almost all prior studies measure only phenological change and not the accompanying change in the light window for the understory, but see [4, 12]. Only local, and not regional or global, studies can test that assumption. Very few studies quantify light availability when phenological responses contrast between co-occurring understory species and the forest canopy strata [3, 4]; also see methods in [30]. Rarely do studies quantify carbon gain of understory species and how it is affected by access to light in spring, summer, and fall [4, 5]. Carbon gain/loss due to relative changes in canopy vs. understory phenology has been estimated to be a loss for herb species by a comparison between the mid-1800s and the present [24] and also by two models that forecast major carbon losses for herbs [24] and tree seedlings [6]. However, to our knowledge, no long-term, continuous community-level study of both strata co-occurring in the same forest has been available to determine trends over time in the extent and direction of phenological asynchrony between strata and understory light and carbon gain/loss, but see [4].

The light-interception consequence of using either date of phenological events or temperature sensitivity as the measure of change depends on when they happen, because light availability increases over spring and decreases over autumn. Comparisons of temperature sensitivity are problematic because, without knowing the specific plant activity dates, it is not possible to know how much total light the understory gains prior to canopy closure or after canopy opening. Regionally-based studies do not necessarily include co-occurring, contrasting strata, so the context of any detected differences between forest strata is missing. Studies at the local level ideally take tree species composition above specific individual understory plants into account, because variation in phenology among canopy tree species affects understory phenology, growth, and reproduction [5, 31, 32]. Ideally, seasonal net carbon assimilation during an understory species' periods in light and shade is available to evaluate net carbon gain over time [4, 10]. Studies show mixed results on the relative importance of photosynthesis and respiration occurring during leaf expansion of trees [33, 34] and find it to be species-specific.

Ultimately, long-term and forward-looking assessments of how changes in mismatches affect plant fitness are desirable [5].

No previous study has directly tested the mismatch hypothesis within one plant community by quantifying phenology of most species of both forest strata within one site. Neither has any study evaluated light availability and temperature effects (in addition to phenology) on understory light limitations, evaluated whether trends are evident in the direction and extent of mismatch over a continuous period of multiple decades with warming temperatures, nor estimated carbon gain/loss to the understory species.

This study aims to evaluate the phenological mismatch hypothesis based not on changes in absolute dates or temperature sensitivities, but on local understory light availability calculated from daily solar radiation measurements, interference by tree trunks/branches and canopy leaves, and *in situ* understory light transmittance. We quantify the relative importance of four potentially limiting factors (understory phenology, suitably warm temperature for photosynthesis, canopy phenology, and solar radiation) on light availability for each understory species over more than two decades. We are unaware of any previous study that has separately quantified how multiple factors individually affect understory plants' light availability. For each species, we compare limitations by these four limiting factors and examine temporal trends for their importance throughout the long study period. Finally, for four herb species with published photosynthetic data, we estimate the carbon consequences from mismatches between their phenology and canopy phenology over the two-decade study period.

A mismatch detrimental to understory plants is demonstrated when understory light availability shifts to be limited more by canopy phenology than understory phenology, after accounting for changes in temperature and solar radiation (Scenario 3 in Table 1). This study also evaluates the importance of the four limiting factors as a function of spring temperature in addition to a simple trend over time. A mismatch detrimental to understory species is demonstrated if light interception decreases in warmer years due to any combination of these four factors (Scenarios 3 and 5 in Table 1), but primarily because of canopy and understory phenology having less of a gap between them (Scenario 3). Cold spring temperatures (Scenarios 4 and

**Table 1. Some scenarios of possible consequences for understory plants due to different combinations of changes in the four limiting scenarios.**

| Scenario number | Type of change in the four limiting factors | | | | Consequence for understory plants |
|---|---|---|---|---|---|
| | Understory phenology | Limiting low temperatures | Canopy phenology | Solar radiation | |
| 1 | Moderately earlier | Little change | Moderately earlier | Little change | Little difference in light understory interception |
| 2 | Much earlier | Little change | Slightly earlier | Little change | Greatly increased light to understory plants |
| 3 | Slightly earlier | Little change | Much earlier | Little change | Greatly decreased light to understory plants |
| 4 | Moderately earlier | Less common due to warmer temperatures | Moderately earlier | Little change | Slight increase in usable understory light |
| 5 | Moderately earlier | More common due to more variable spring temperatures | Moderately earlier | Little change | Slight decrease in usable understory light |
| 6 | Moderately earlier | Little change | Moderately earlier | Increased due to less cloud cover in spring | Slight increase in understory light to plants |

This table focuses primarily on the spring because it is more important than the autumn for most understory species' light budgets. It does not cover all possible combinations of changes in these four factors. These six scenarios highlight different key outcomes. Elements of more than one scenario may combine in some cases.

5 in Table 1) or changes in solar radiation due to cloud cover (Scenario 6) are also possible, but likely of lesser magnitude.

## Methods

Because of the complexity of the analyses presented here, we first outline how the different parts of this study's methods fit together, and then give detailed descriptions of each component. The individual components are numbered so the big picture given here can easily be related to the component sections and the overview in Fig 1.

This study builds on several (1) primary data sources from the vicinity of Urbana, Illinois, USA. Phenology dates of (1a) canopy trees [35] and (1b) understory saplings [35] and herbs [36], and (1c) percent canopy light transmittance were collected by one of us (CKA). The (1d) species' basal areas (cm$^2$) came from a 2005 survey of all trees in the study site, with modifications due to extensive death of ash trees from 2016 onward. Environmental data on (1e) temperature (˚C) and (1f) solar radiation (W/m$^2$) came from the National Weather Service and NOAA Surface Radiation Monitoring (SURFRAD), respectively. The canopy light transmittance, basal area and canopy phenology datasets were used to build a (2) model of daily canopy light transmittance percentage which accounts for both (2a) light interception by branches and trunks, and (2b) light interception by canopy leaves. This model, in combination with above-canopy solar radiation and (3) estimations of understory plant photosynthetic leaf area, was combined with understory phenology and temperature data to calculate the (4) species-specific estimated annual light interception of saplings and herbs. The deviation from the long-term mean of light interception was (5) partitioned among four potential causes: Understory phenology, cold Temperatures, Canopy phenology, and solar Radiation–the uppercase letters $U$, $T$, $C$ and $R$ are used throughout the text as abbreviations for these factors. Finally, long-term trends in the partitioned light estimates as a function of both year and spring temperature were analyzed using (6) linear regression, and photosynthetic consequences of phenological changes were computed for a selection of herb species. The interrelationships among the elements summarized here are shown graphically in Fig 1.

### 1 Data sources

**Study site and species.**   The study site is the northern half of Trelease Woods, a 24-ha deciduous old growth forest fragment located 5 km NE of Urbana, IL, USA (40.13˚ N, 88.14˚ W). Field site access was given by Steve Buck, Natural Areas Manager, University of Illinois. Its topography is level, with elevation varying by < 5 m. Among the 20 canopy tree species, the dominant species by basal area during the study were *Acer saccharum* Marsh., *Celtis occidentalis* L., and *Fraxinus americana* L., although *Agrilus planipennis*, the emerald ash borer, nearly extirpated all *Fraxinus* individuals from the forest by the end of the study period. Saplings of canopy species are dominated numerically by *Acer saccharum* (sugar maple) and *Aesculus glabra* Willd. (Ohio buckeye). The herb community is species-rich and phenologically diverse, with some species having leaves at all times of the year. Most species emerge in spring with completion of leaf expansion and flowering occurring at or soon after canopy closure. Species senesce and become dormant in varying seasons: early summer for spring ephemeral species, summer or autumn for other species. Some species retain leaves through the winter.

**1a Canopy tree and sapling phenology.**   A total of 187 individual canopy trees were included in the phenological censuses. They were located haphazardly in the study area and were selected to include the 15 most common species which together represented 96.3% of total basal area of trees with a DBH ≥ 22.9 cm (9") in 2005 (S1 Table). Each species was represented by a mean of 12 trees (range = 1–20; S1 Table).

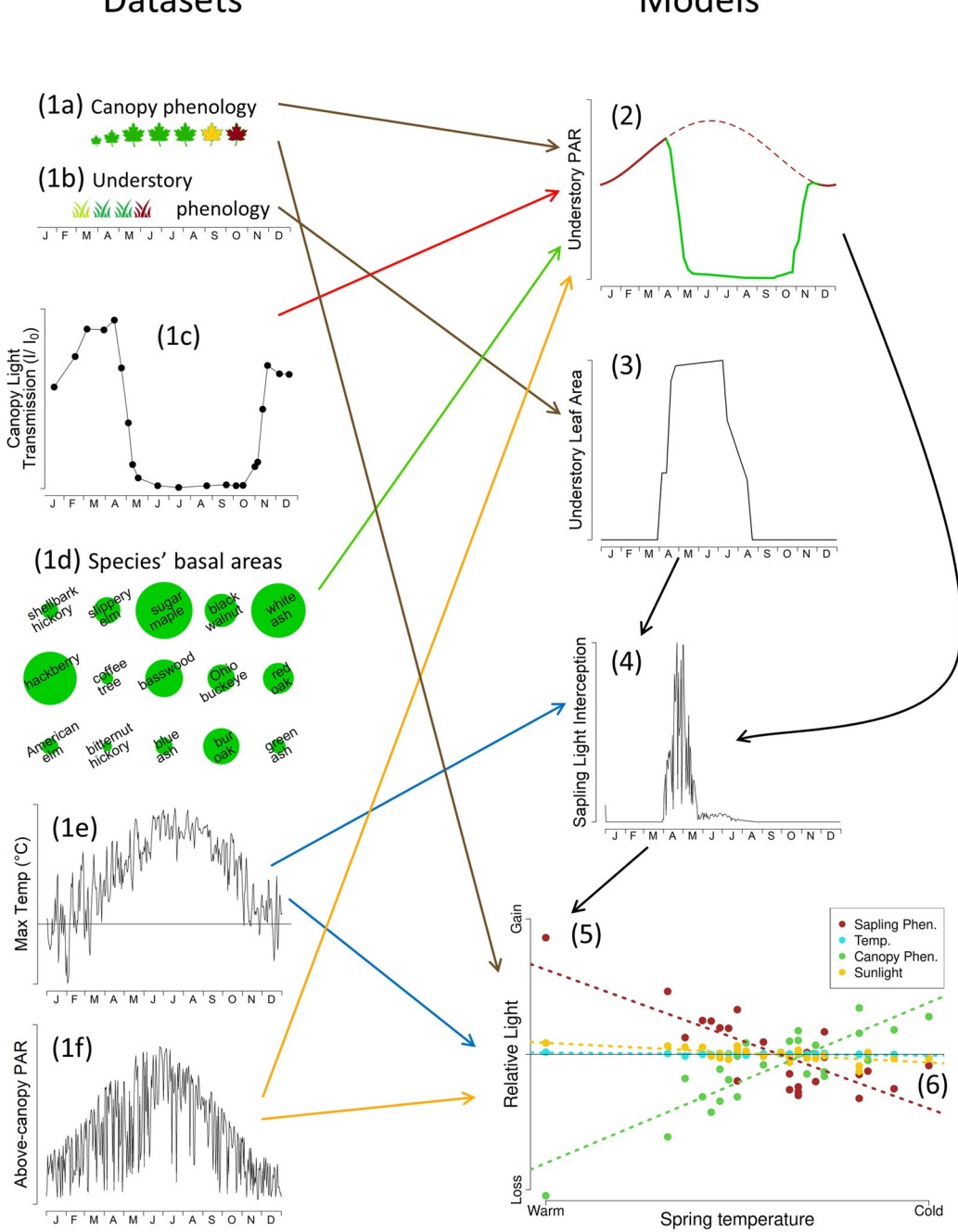

**Fig 1. A conceptual diagram of this study's analysis to complement the overview given in the second paragraph of the methods.** For clarity, Section 1 shows hypothetical data for a canopy tree (1a) and a spring ephemeral herb (1b), Sections 1d-f and 2–4 show only one of many possible years (1995–2017 for herbs and 1995–2022 for saplings) of data or model outputs, and Sections 3–5 show only one sapling species. The herb and sapling analyses follow the same general pattern, although with more complications due to phenological complexity of some herb species (see Methods: Section 3).

Trees were observed on the same day of the week by the same observer (CKA) from 1995–2022. At 7-day intervals during spring and autumn, binoculars (8x magnification) were used to determine the phenological status of each individual tree or sapling. All observations were

of phenological stages (i.e., the status of a plant on a particular date) rather than phenological events (i.e., the date on which a particular stage was reached). All observations and analyses reported here take a stage- rather than event-focused perspective. Seven possible stages were noted in the spring: no extension (<E1: buds in winter state or swelling, but no growing tissue has emerged), three stages for bud burst, shoot extension and leaf unfolding (E1-E3), and three stages for leaf expansion (F1-F3). In the autumn trees were assessed as fully leafed (<D1) or in one of three stages of leaf drop (D1-D3). The three stages of a phenological phase (e.g., E1, E2, E3) correspond to one-third, two-thirds, and the entire phase being completed; see S2 Table for details of each stage's designation. Bud swell (with no protrusion of shoot/leaf parts) and senescence (leaf coloration) were not included in our calculations because initial analyses showed they had no discernible impact on canopy light transmission.

The raw phenology observations for trees were organized to list the first census date when each phenological stage was observed for each individual tree in each year. If more than one stage was completed between census dates, the skipped stage(s) had no corresponding date. For the purpose of analysis, these observations were rearranged to list the current phenological stage of each tree on each census date. Thus, for each census that predates any phenological activity by a tree in a given year, we assigned a phenological stage of <E1. Similarly, all dates after fully expanded leaves were observed were assigned a stage of F3. An analogous process (but in the opposite direction) was applied to autumn data.

**1b Understory phenology.** A total of 36 saplings were included in the phenological censuses. The tree saplings included sugar maple (n = 11) and Ohio buckeye (n = 10), which together represented 73% of individuals in the 7.6–15.2 cm DBH size class in 2005. Also observed were saplings of *Fraxinus quadrangulata* Michx. (blue ash; n = 15 at start of study with all but one dying by 2022). The saplings were observed on the same dates as canopy trees. The phenological stages used for sapling observations were the same as for canopy trees.

Herb phenology observations were nearly always made on the same dates as canopy trees and saplings. Herbs of 33 species were observed, including two distinct cohorts of leaves in certain perennial species or 1st vs. 2nd year individuals in a biennial species, for a total of 37 species-cohort combinations. Data requirements (see following paragraph) led us to drop four of the combinations for a final total of 33 analyzed (see S3 Table). As the cohorts within a species had contrasting phenologies, they were analyzed separately; we hereafter refer to these species-cohort combinations as "species" for simplicity. The observations were made in 25 1-m$^2$ square plots evenly arranged on a 50 x 50 m grid beginning 50 m from any forest edge. The predominant phenological stage of each herb species in each plot was determined weekly in spring and summer, biweekly in fall, and monthly in winter from 1995–2017 (see [8]). Observations stopped after 2017 because many herbs' populations had diminished substantially [37]. For each herb species in each plot, field observations were made of dates of emergence (Em), end of shoot/leaf expansion (Ex), beginning of senescence (leaf coloration - Se), and dormancy (disappearance of above-ground structures - Do). In addition, for *Cardamine douglassii*, the date of cotyledon emergence was observed.

Many species had certain phenological stages that were difficult or impossible to observe, so species-specific data-cleaning rules were followed to fill in missing values or omit observations for a year/species/plot combination (see S1 File). Following cleaning, species with at least 10 complete years of observations spanning at least 15 years were analyzed as described in the following sections. For some species, the number of plots observed (or which particular plots had observed plants of that species) varied among the years, but species were used as long as they met the requirements described above. The final list of 33 species and their phenological seasonality is in S3 Table.

**1c Canopy light transmittance.** Mid-day light transmission observations were collected on 21 clear sunny days throughout 2002, with weekly measurements made during the spring and autumn periods when canopy status was changing quickly. Instantaneous readings of irradiance were made with a quantum sensor (LI-190R, Li-Cor, Lincoln, Nebraska, USA) to calculate the proportion of irradiance transmitted to the understory ($I/I_o$, where $I$ = irradiance (W/m$^2$) reaching understory individuals and $I_o$ = irradiance in the open). $I_o$ readings were made in an open field 25 m west of the forest edge at the beginning and end of each 45-minute sampling period; $I_o$ was the mean of the two readings. Irradiance ($I$) was measured at 50 haphazardly-selected sites by holding the sensor horizontally at a height of 1.4 m in areas not experiencing sunflecks or overtopped by nearby leaves. The mean value of the 50 readings was used to calculate $I/I_o$ for each date.

**1d Species' basal areas.** A complete census of the northern half of Trelease Woods (where the phenology observations were made) was undertaken by J. Edgington in 2005. Species' basal areas (cm$^2$), aggregated to per-species proportions of total study-forest-wide basal area, were calculated for stems larger than 22.9 cm (9") DBH measured at 1.4 m above the soil surface (S4 Table). Following [38], subsequent calculations assume that species' leaf areas are proportional to their basal areas.

**1e Temperature.** Daily maximum and minimum temperatures (˚C) from 1995–2022 were recorded at the Champaign, IL Weather Station (3S), 8 km SW of the study site. The station is part of the National Weather Service Cooperative Observer Program (US-COOP, www.weather.gov/coop/) and its data can be downloaded from the Midwestern Regional Climate Center (mrcc.purdue.edu/). The open area surrounding the station is similar to the edge of the study site, resulting in temperatures slightly warmer than those experienced by study plants in the forest interior.

Because leaves do not reliably photosynthesize at low temperatures, we used the temperature data to apply certain restrictions on which days' light interception values were accumulated toward seasonal/yearly totals, based on daily maximum temperature and solar radiation values. For the 0.09% of days missing maximum temperature readings, we filled in the missing data with the average value for that Julian calendar date from 1995–2022. Each day was assigned a temperature limitation value ($T_{d,y}$–see Section 4 below) of either 0 (data not used) or 1 (data used) based on the following rules:

1. If the daily maximum temperature was > 5˚C, $T_{d,y}$ = 1.

2. If the daily maximum temperature was ≤ 0˚C, $T_{d,y}$ = 0.

3. If the daily maximum temperature was between 0˚C and 5˚C, $T_{d,y}$ = 1 if the day was sunny and otherwise $T_{d,y}$ = 0. We declared a day to be sunny if that day's PAR value at nearby Bondville, IL (see Section 1f below) was at least as high as the median PAR observed across all years (1995–2022) within ±3 days of that Julian date.

4. If the daily maximum temperature was between 0˚C and 5˚C and the PAR measurement was missing (conditions that occurred on only 0.02% of all days from 1995–2022), daily light interception values were randomly chosen for inclusion ($T_{d,y}$ = 1) or exclusion ($T_{d,y}$ = 0) with the same probabilities of inclusion/exclusion for all days with temperatures between 0˚C and 5˚C.

**1f Solar radiation.** The geographically-closest source for daily total photosynthetically active radiation (PAR) above the canopy level was the NOAA surface radiation monitoring (SURFRAD) site at Bondville, IL, 19 km SW of the study site (gml.noaa.gov/aftp/data/

radiation/surfrad/Bondville_IL/). The readings are given in units of energy per area (W/m$^2$) and were reported over 3 min intervals from 1995–2008, and over 1 min intervals from 2009–2022. We multiplied the raw values by their respective time interval and summed over the entire day to estimate total daily incoming PAR in energy/area units (W min/m$^2$). Similar to the temperature readings, certain missing values were replaced by mean values for that particular Julian calendar date over 1995–2022. Values were determined to be missing if (1) no data file was available for a given day, (2) the daily data file was incomplete, (3) the values were unreasonably dim, which we defined as a maximum value of $\leq 10$ W/m$^2$, or (4) $\geq 5\%$ of values were internally flagged as failing quality control (QC). Of the days from 1995–2022, 7.85% of light readings were missing, particularly due to a $\sim 1$ year period from May 2001 to May 2002 in which all PAR values were flagged as having failed QC.

## 2 Estimation of daily understory light availability

This section explains how we estimated the percentage of sunlight that was intercepted by the forest canopy before reaching the understory. The canopy light interception model has two parts: light interception by stems and branches (Section 2a) that are present year-round, and light interception by canopy leaves that are seasonally present (Section 2b). We parameterized our model for both of these phenomena using the mid-day canopy light interception ($I/I_o$) data, with a particular focus on the periods of spring and autumn when the canopy was changing the quickest.

For each day from 1995–2022 we estimated mid-day canopy light transmittance as a function of solar angle and phenology. For the periods before any tree began budburst (stages $< E1$) and after all trees completed leaf drop (D3), we simply used the modeled light transmission curve due to interception by trunks and branches alone (Section 2a; solid brown line in part 2 of Fig 1). During the period from the end of spring green-up to the beginning of autumn leaf drop, daily estimated light transmission also included a component based on leaf phenology (Section 2b; green line in part 2 of Fig 1). Values on non-census dates were interpolated linearly between census dates. For the summer period when all trees had fully-expanded leaves (F3), we interpolated between the end of spring green-up and the beginning of autumn leaf drop.

**2a Model of branch and trunk light transmission.** Here we explain how the amount of light intercepted by tree structures present throughout the year (branches and trunks) was estimated. Two approaches are possible to model how much light passes through tree trunks and branches to reach the forest understory. The more fundamental approach incorporates the path length of the sun through the canopy, which can be computed from solar geometry based on the latitude of the study site and day of the year. We tried this approach, but it was not possible to align the results well with the observed canopy transmittance data. This is likely due in part to the non-random orientation of tree trunks and branches. Instead, we used an approach of fitting a function with a minimum set of assumptions to the observed canopy transmittance values for the parts of the year when no leaves were present. While a previous study found no clear sign of attenuation of light in a leafless deciduous canopy in a lower-stature forest (mean canopy height 21.5 m compared to 28 m in Trelease Woods) at a somewhat lower latitude (36° N compared to 40° N) [2], our $I/I_o$ observations show a clear pattern related to seasonal solar angle during the non-leafy period (see black dots in Fig 2). Thus, we chose to fit a canopy transmittance function with the following properties:

1. The maximum value over the year period is reached on the summer solstice,

2. The minimum value is reached on the winter solstice,

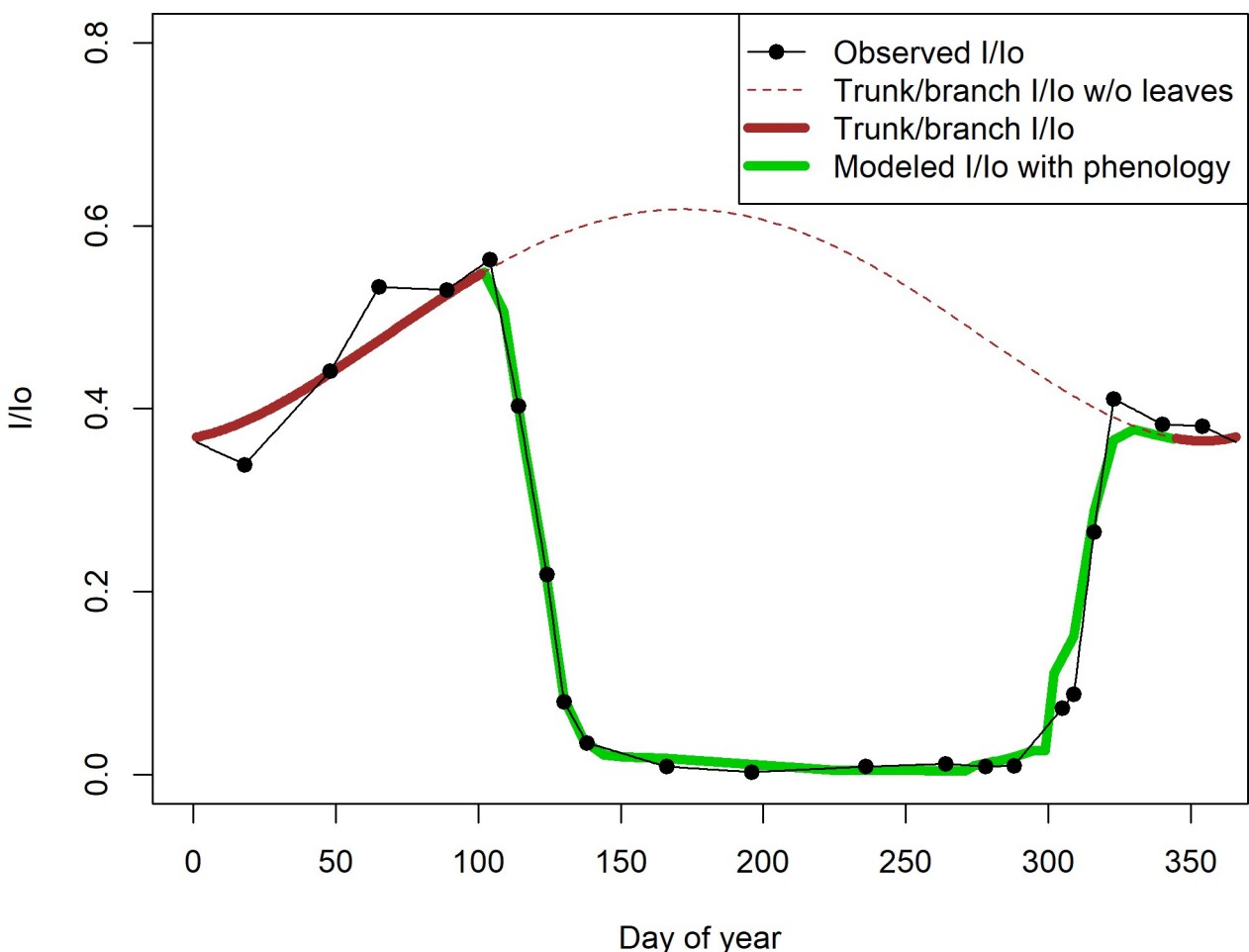

**Fig 2. Canopy light transmission in Trelease Woods in 2002.** The black dots show observed ratio between understory irradiance and irradiance in a nearby open area on clear days in 2002 ($I/I_0$). The brown lines (solid and dashed) show modeled light transmittance by trunks and branches in the leafless forest (see Methods: Section 2). The green line shows canopy light transmittance as modeled using the trunk and branch model, plus canopy leafing phenology data. Thus, the brown and green thick solid lines together show the entire year's course of modeled understory light. Calibrations from these 2002 observations were used to estimate daily understory light in all study years (see Methods: Section 2).

3. There are no local maxima or minima at other times of the year, and

4. Symmetry around the summer solstice.

These criteria can be met with a strictly even 4th degree polynomial

$$T_C(D) = \beta_4 D^4 + \beta_2 D^2 + \beta_0$$

where $\beta_4$, $\beta_2$ and $\beta_0$ are fitted parameters, $T_C$ is the canopy light transmittance, and $D$ is the day of the year. To simplify the process, we use a modified calendar with the summer solstice on $D = 0$ and the winter solstice on $D = \pm 183$ (values beyond $D = \pm 183$ are part of a different solar year, so were not used in this analysis). Criterion (2) restricts the derivative of $T_C$

$$\frac{dT_C}{dD} = 4\beta_4 D^3 + 2\beta_2 D$$

to be zero on the winter solstice (i.e., when $D = 183$), so that

$$4\beta_4 183^3 + 2\beta_2 183 = 0.$$

This restriction means that $\beta_4$ and $\beta_2$ have a fixed ratio such that:

$$\beta_2 = \frac{-4*183^3}{2*183}\beta_4 = -66978\beta_4$$

Substituting this into the original polynomial reduces its total number of parameters to two ($\beta_4$ and $\beta_0$):

$$T_C(D) = \beta_4(D^4 - 66978D^2) + \beta_0$$

To estimate values of the parameters $\beta_4$ and $\beta_0$, we first linearly interpolated $I/I_o$ values between all observed dates, including a wrap-around between the last and first observed $I/I_o$ dates of the year. We then calculated $D^4$–66978 $D^2$ for Julian dates 1 to 113 and 334 to 365 (the non-leafy period in 2002) and fit an ordinary least squares model between these values and the interpolated $I/I_o$ values using the lm() function in R version 4.2.0 [39]. Values from leafy times of the year (days 114–333 in 2002) could not be incorporated because data on branch/trunk interception alone cannot be collected. This gap in the data has little effect on the overall calculations because understory light levels in the summer are overwhelmingly a function of canopy leafiness (green line in Fig 2).

**2b Model of light transmission by canopy leaves.** In this section we explain how the seasonally changing percentage of light intercepted by canopy leaves was calculated (the thick green line in Fig 2). Canopy light transmission considering leaf phenology must be modeled with respect to the trunk and branch light-interception model derived in the previous section. This separate leaf interception model is necessary because canopy transmission measurements were made only on selected dates during 2002, and we needed estimates of this variable for all dates from 1995–2022. This is a complex process whose description fills the remainder of this section.

The first step is to estimate canopy transmittance values (on a scale of 0 to 1) for the observed phenological stages E1 . . . F3. The numerical values of E1 . . . F3 were then used to compute the estimated forest-level canopy transmittance on each census date. To do this, we first calculated the species-level phenological stage by converting each tree's phenological stage to a transmittance value (see following paragraphs for explanation of this process) and averaging these. We then calculated the forest-level canopy transmittance by averaging the species' values, weighted by species' relative basal areas (dataset 1d in Fig 1), and multiplying this value by the census date's trunk and branch transmittance (the dashed brown line in Fig 2). When repeated on each spring census date for which leaves were present, this process yields the downswing portion (from approximately day 110 to day 195) of the green curve in Fig 2. A similar process was followed for the autumn upswing in canopy transmittance (the right side of the green curve in Fig 2). Finally, transmittance values for the summer period when no phenological changes were taking place were filled in by linear interpolation, completing the thick green curve in Fig 2. Note that in the following paragraphs light interception is equivalent to 1 - transmittance, so transmittance = 1 - interception.

The spring leaf phenology observations from 1995–2022 can be related to canopy light interception by assigning a numerical value to each phenological stage. There is no way to do this from first principles–the phenological stages were defined with respect to how leaves look to an observer on the ground, not regarding their light-interception ability. Thus, we have to

estimate each stage's relative contribution to light interception. This process can be simplified by applying a few basic restrictions:

1. Canopy phenology stages before E1 have a value of zero; bud swelling has a minimal impact on light,

2. Subsequent stages have increasing contributions to light interception, so E1 < E2 < E3 < F1 < F2 < F3, and

3. F3 can be < 1 because even fully-expanded leaves do not intercept 100% of light.

The unknown values of E1 . . . F3 are a six-dimensional optimization problem, so finding precise values that optimize the fit between observed and estimated canopy-level light interception for each stage is computationally prohibitive. Instead, we determined near-optimal light interception values for each stage E1-F3 using an iterative method to explore the complex parameter space. First, each stage was assigned an initial light-interception value between 0 and 1 with equal spacing among stages, so E1 = 1/6, E2 = 2/6 . . . F3 = 6/6, chosen as an easy-to-calculate starting point for the iterative optimization (see following paragraph) that conforms to the three rules listed above. The mean phenological stage on each census date in 2002 was calculated for each species. These values were aggregated to the community-level canopy light interception by leaves ($L_c$) by taking means weighted by each species' proportional study-site level basal area (see Section 1d). Then, a value of $I/I_o$ was computed by multiplying the modeled light transmission value for the leafless canopy by 1 - $L_c$. Daily $I/I_0$ estimates were linearly interpolated between the phenology census dates. We calculated the difference between these estimated values and the observed $I/I_o$ values (Section 1c) on all measurement dates in 2002 during the green-up period. The sum of the squares of these error values was used as a measure of model error.

Next, we repeated this procedure 10,000 times, each time changing every phenological stage's light-interception value by a randomly-generated amount. The random jumps were generated from a Cauchy distribution (similar to a normal distribution but with a greater probability of occasional very large jumps) with a variance = 1 / (step number), so average jump size decreased as the algorithm proceeded. Any jumps that resulted in values not meeting the three restrictions listed above were rejected, and new randomizations were performed until an acceptable set of values was reached. With these new phenological values, the species- and canopy-level aggregations were repeated, and new predicted values of $I/I_o$ were generated. If the sum of squared errors between these values and the observations was lower than in the previous step, then these phenological parameters were accepted as the new provisional values, and used as the starting point for subsequent jumps until a better set was discovered. This process, analogous to simulated annealing [40], is an efficient way to reach near-optimal parameter values when the space has too-high dimensionality to fully explore and is too jagged to use traditional methods for finding absolute optima. Because this process is stochastic, it does not give the same result every time. However, the parameter estimates found by different runs of this process were so similar that differences among them had no discernible impact on our results.

Similar to the springtime green-up, the autumn brown-down can be related to canopy light transmission by estimating a numerical value for each autumn phenological stage. We first tried modeling this process as the product of two separate processes, leaf coloration (senescence) and leaf drop. The logic was that the amount of light intercepted by the canopy is proportional to how many leaves are left, and also that the light blocked by leaves depended on the extent they had senesced. However, our initial attempts to fit stage-specific parameters indicated that model fit was best when senescence led to essentially no reduction in light

blocking by leaves. For this reason, we simplified the model to include only the leaf-drop process. As in the spring, we placed certain basic restrictions on the light-transmission value at different stages:

1. F3 ≤ 1 since even fully-expanded leaves do not intercept 100% of light, and

2. Subsequent stages have decreasing contributions to light interception, so F3 > D1 > D2 > D3, with D3 fixed at 0 because non-existent leaves don't intercept light (as in spring stages < E1); we tried allowing D3 to vary freely, and it always converged toward 0.

We estimated parameter values for these stages using the same type of iterative procedure described for the spring data. The fit was not as clean as in the spring–note the jagged steps in the autumn curve for 2002 (Fig 2); this is probably due to the autumn brown-down being represented by fewer phenological stages than in the spring green-up.

## 3 Estimation of understory plants' leaf display

This section outlines the method we used to estimate relative seasonal leaf potential for light interception by saplings and herbs. Understory plants' leaf display places a fundamental limit on how much light they can intercept. If a plant grows leaves relatively early or late, this has important consequences for how much of the sunlight reaching the understory they can absorb. We compute these values on a relative scale of 0 to 1 because total leaf areas were impractical to measure, and our ultimate interest is in understanding the relative impact of phenological shifts on species' light interception, not leaf area as such. Therefore, although we refer to this variable as "leaf area" for the sake of simplicity, it also encompasses another aspect of leaf function, namely the amount of light-absorbing pigment present. Because the canopy calculations estimate overall light interception, they encompass both leaf area and pigmentation. That saplings follow a similar light-interception trajectory to canopy trees is an assumption that we make at this point, whose implications we address in the discussion section. The remainder of this section explains how these relative values (model 3 in Fig 1) are derived from the basic phenological observations (dataset 1b in Fig 1).

**Saplings.**   For each of the three sapling species (sugar maple, Ohio buckeye, and blue ash), we calculated the relative amount of leaf area available for photosynthesis on each date of each year from 1995–2022. Note that we are not attempting to calculate the total amount of light intercepted by understory plants, but the amount per unit of total leaf area, which varies greatly among individual plants and species. For the spring, these calculations were based on phenological stages. First, we made a list of phenological observation dates for each species in each year and filled in the phenological stage observed on that date. If no change had been observed, then the stage was taken to be the same as the previous observation. The phenological stages then were replaced by the corresponding numerical leaf expansion estimates from Section 2b, but re-scaled to range from 0 to 1 by dividing by the maximum value (by definition the interception value of stage F3). A value of zero was inserted for the date(s) on which stages <E1 were observed.

For the autumn, a slightly different approach was used because, while leaf drop in the canopy is a good determinant of light transmittance, senescence in understory leaves comes with declining photosynthetic capacity [12]. Thus, we used a linear decline with phenological stage (F3 = 1, S1 = .667, S2 = .333, S3 = 0) to estimate light interception ability by saplings. While these choices are the simplest possible assumption about autumn light interception, in practice, the details of these assumptions make little difference to our subsequent calculations because the understory is so dark late in the year when the saplings' leaves are senescing.

If a plant passed through a phenological stage without it being observed (which was common with S1), the following steps were taken to fill in the gaps. If there was no observed date with stage S1, then a value of 1 was assigned to the census date prior to the first observation of a later S stage. For a few early-autumn senescence observations (particularly for Ohio buckeye), there was no previous autumn census date for that year (i.e., senescence was observed at the first autumn census). In this case, a value of 1 was assigned seven days before the first autumn census date. An analogous approach was applied to determining the date that a leaf proportion of 0 was reached. Observations of Ohio buckeye saplings' leaf drop, which happens in mid to late summer, were entirely missing in 2019 due to CKA's travel schedule. In this case, all Ohio buckeye sapling leaf area proportions for 2019 were set to NA, effectively eliminating those saplings' data for that year.

Dates between censuses were filled in by linear interpolation. This interpolation procedure resulted in all saplings' proportional leaf areas having a valley with a value of 0 in the winter, and a plateau with a value of 1 in the summer. The spring and autumn periods were increasing or decreasing functions, respectively, although sometimes with brief step-like pauses.

**Herbs.** We performed leaf display seasonality calculations similar to those for the saplings for most of the herb species present in the understory of Trelease Woods. However, herbs presented several complexities beyond those faced in sapling calculations. These include:

1. Phenological complexity–Unlike the studied sapling species which are strictly deciduous and vary only a few weeks among years in their dates of leaf growth and loss of green coloration, many of the herb species have more complicated patterns, including biennial, evergreen or wintergreen leaves, more than one flush of leaves per year (S3 Table), and slow and indistinct senescence. These challenges were addressed for each species as detailed in S1 File.

2. Census methods–For the sapling calculations, the plants were tagged, ensuring that the exact same individuals were observed every year. Because most of these (mostly perennial) herb species have no stems/leaves that persist above the leaf litter during part of the year, tagging plants is impractical. Thus, observations in each census were average values for all plants of a species found within a fixed 1-m$^2$ plot.

3. Rarity and declining populations–Some of the herb species found in Trelease Woods were always rare. Other species declined substantially during the study period [37]. This meant that in some years of the study, no observation of some species was possible. This limited inclusion of some species in this study, resulting in the exclusion of a few species as discussed in Section 1b.

4. Missing observations–In many cases, due in part to leaf litter, certain stages of herb phenology were not observed in certain years, even though it can be logically deduced that they happened. For example, if newly-expanded leaves are observed, the plant must clearly have emerged from dormancy at some point, but those dates were sometimes missed. To estimate missed dates like these, or decide to drop a plot-year-species combination with too many missing values, we followed a set of rules explained in S1 File.

The general procedure for converting herb observations to the seasonal relative leaf area values ranging from 0 to 1 was similar to that for saplings. However, there were many species with special circumstances (e.g., cotyledons or semi-evergreen phenology) that had to be accounted for. This procedure and species-specific exceptions to it are described in the section of S1 File titled "Estimation of understory plants' relative leaf area."

## 4 Estimation of species-specific understory light interception

This section explains how we estimate the impact of understory phenology, cold temperatures, canopy phenology and sunlight on understory plants' light availability. Light interception is a multiplicative process. By this, we mean that the canopy absorbs or reflects a percentage of above-canopy light that changes over time with solar angle and leaf phenology as calculated in Section 2, and saplings and herbs absorb some amount of what is transmitted by the canopy that depends in turn on their own leafing phenology (Section 3). The absolute amount of usable light absorbed by saplings and herbs depends on four factors of interest. In order of decreasing rank (explained in Section 5), they are Understory phenology, cold Temperatures, Canopy phenology, and solar Radiation (abbreviated $U$, $T$, $C$ and $R$). The relative amount of light harvested by an understory plant is the product of these four variables, where the subscripts $d$ and $y$ indicate Julian calendar date and year:

$U_{d,y}$–The phenological status of the understory plant (see Section 3 above). This is essentially the area (relative to full leaf area) of leaves displayed by a plant on a particular day. This variable ranges seasonally from 0 (no leaves displayed) to 1 (leaves are fully expanded and have not yet begun to senesce).

$T_{d,y}$–Physiological suitability of air temperature for photosynthesis. This variable takes a value of 1 if the daily maximum air temperature is $> 5$ ˚C, or if the temperature is between 0 ˚C and 5 ˚C and the day is sunny (see Section 1e above for more details on these calculations); otherwise, this variable has a value of 0. These values are broadly in line with physiological measurements for some herb species, although photosynthetic temperature responses of understory herbs acclimate to changes in temperature over the course of a day or so [41, 42]. Because of uncertainty and complexity in real-world photosynthetic temperature responses, our inference on this factor gives a general indication of how limiting cold temperatures may be for the studied species.

$C_{d,y}$–The forest canopy's transmittance of light, in other words, the proportion of incident light that is not intercepted by branches or leaves in the canopy (see Section 2 above). Branch interception varies seasonally due to more light hitting branches before reaching the understory when the sun is at a lower angle (see Section 2a). Leaf interception varies seasonally due to canopy tree phenology. In theory, this variable ranges from 0 to 1, but in practice our study site has a minimum value around .02 (in summer when the canopy leaves block most light) and a maximum value of about .55 (in late spring, when the sun is relatively high in the sky but leaves have not yet appeared). This value never comes close to 1 because so much light is blocked by branches and trunks, especially in the winter when the sun is relatively low in the sky.

$R_{d,y}$–The amount of photosynthetically active radiation (PAR) reaching the forest canopy (see Section 1f above). This variable increases seasonally with increasing solar angle and day length and varies daily due to changes in atmospheric transmittance of light, mostly due to cloudiness. This variable has units of W min/m$^2$ (see Methods Section 1f).

For each day in 1995–2022 we estimated the amount of intercepted understory light by taking the product of $U_{d,y}$, $T_{d,y}$, $C_{d,y}$, and $R_{d,y}$. Because $U_{d,y}$, $T_{d,y}$, and $C_{d,y}$ are unitless, the resulting values have the same units as $R_{d,y}$ (W min/m$^2$). These values can be thought of as understory plant light interception relative to maximum leaf area–that is to say, if displayed leaf area were 50% of its maximum value due to either partial expansion or partial senescence of leaves, then interception would be reduced to half of the value of light available (per m$^2$). This makes the units a bit unintuitive to interpret. They are essentially relative light interception, with relative

being operative only within a species, as it is relative to total leaf area displayed by that species. Thus, we focus on their relative values and do not report units as these are not meaningfully interpretable. The calculated values are useful for comparing the impact of the four limiting factors within a species, and also for examining temporal trends within a species in the importance of these factors. We recommend interpreting differences among species in the relative sizes of the limiting factors cautiously. They can be used to indicate the variability within species, and thus relative importance of the different factors to individual species.

## 5 Partitioning of understory plant light gain/loss among the four limiting factors

This section explains how the relative gain or loss of light available to understory plants is partitioned among the four factors outlined in the previous section. A change in any one of the four variables used in Section 4 affects the impact of the other three variables. Thus, calculating the impact of a change in one variable requires partitioning its interaction with the other three variables, as is described in the remainder of this section. The logic of this partitioning is shown geometrically in the figure in S2 File, although that figure shows only two of the four variables to make it practical to depict.

The ultimate goal of these calculations is to estimate the difference between observed and average understory light interception ($\delta L_{d,y} = L_{d,y} - \bar{L}_d$) and to partition the value of $\delta L_{d,y}$ among the four possible causes ($\delta U_{d,y}$, $\delta T_{d,y}$, $\delta C_{d,y}$ and $\delta R_{d,y}$, where $\delta U_{d,y} = U_{d,y} - \bar{U}_d, \delta T_{d,y} = T_{d,y} - \bar{T}_d$, etc.). The amount of light that would be intercepted under average conditions for a particular day of the year can be computed by multiplying the average values of the four variables:

$$\bar{L}_d = \bar{U}_d \bar{T}_d \bar{C}_d \bar{R}_d$$

These averages are calculated across years as $\bar{R}_d = y_d^{-1} \Sigma_y R_{d,y}$ where $y_d$ is the number of years with values of $R_d$ for date $d$; analogous logic applies for the other average variables. The relative amount of light intercepted by a plant in the understory on a particular day ($L_{d,y}$) is similarly the product of these four observed variables (see Section 4):

$$L_{dy} = U_{dy} T_{dy} C_{dy} R_{dy}$$

$\delta L_{d,y}$ is partitioned into contributions from the four factors ($U$, $T$, $C$ and $R$) which we refer to as $\delta L U_{d,y}$, $\delta L T_{d,y}$, $\delta L C_{d,y}$ and $\delta L R_{d,y}$, respectively. However, partitioning $\delta L_{d,y}$ is not simply a matter of proportionally dividing it among the factors $U$, $T$, $C$ and $R$. Rather, there is a hierarchical ordering of the contribution of these variables: $U$ is most important, followed by $T$, $C$ and finally $R$. The logic behind this ordering is that if a plant has no leaves (i.e. $U_{d,y} = 0$), then $\delta L_{d,y}$ is due entirely to variation in $U$; the values of $T$, $C$ and $R$ make no difference whatsoever. Next, if a plant has leaves but it is too cold for photosynthesis, then the remaining part of $\delta L_{d,y}$ is fully due to $T$; the status of the canopy ($C$) or solar radiation ($U$) is irrelevant. Next comes $C$, and finally $R$. When all observed values of $U$, $T$, $C$ and $R$ are below average (i.e. $\delta U_{d,y}$, $\delta T_{d,y}$, $\delta C_{d,y}$ and $\delta R_{d,y}$ are all negative), the partitioned values are calculated sequentially; additional loss of $L$ is due entirely to the next variable in the $UTCR$ hierarchy with no interaction with other terms. Whatever light loss due to $T$ is calculated as a proportion of what is remaining after $U$ has been accounted for, and losses due to $C$ are calculated only after $U$ and $T$ have been accounted for, etc. However, depending on how many of the values of $UTCR$ are positive, then 2-, 3- and/or 4-way interactions must be included. This logic is formalized in the equations presented in S2 File.

For each year, we calculated these values for each individual sapling or species within a plot of herbs (note that $\delta LT_{d,y}$, $\delta LC_{d,y}$ and $\delta LR_{d,y}$ are not the same for all saplings or herb plots within a year because all three depend on $\delta U_{d,y}$ that affects calculation of the other terms). We then summed these values across all days within a year and averaged over all saplings or plots of each species to get species- and year-specific values of light gain/loss due to each of the four factors.

## 6 Statistical analyses of trends in and among limiting factors

This section outlines how trends over the years of the study in the four limiting factors ($U$, $T$, $C$, $R$) were analyzed. We computed the standard deviation of each factor for each species as an indicator of which factors were more variable, and thus most limiting to that species. Note that we do not simply use the mean values of these factors because the partitioning process described in Methods Section 5 leads to them having long-term averages near zero (the deviation from exactly zero is due to the hierarchical relationship among the factors). Therefore, for instance, if understory phenological dates ($U$) vary a lot from one year to another, then the relative gain or loss of light due to that factor will be quite big as long as this is not happening in the middle of the summer; there has to be light to gain or lose for it to have a big impact. If $U$ is the same every year, then there will be no variation in how much light is gained/lost due to that particular variable. Similar but slightly more complex logic applies to the other limiting factors. The additional complexity is due to variation in more fundamental factors reducing the impact of a particular factor. For instance, two hypothetical years of identical canopy phenology would still have different values for canopy limitation if understory plant phenology is different, thus affecting how much canopy phenology actually matters. Based on these calculated values for the four limiting factors, we computed ordinary least squares regression for the trend over years of the study time for each factor-species combination. In all cases we used a p-value of $\leq .05$ as the threshold for describing a result as statistically significant. We also did this regression for the sum of the $U$ and $C$ (the two phenology factors) and the sum of all four factors as indicators of the relative change in that species' total light interception.

Our next analysis looked at the limiting factors as a function of annual spring temperature. The simplest way to compute this variable is mean temperature over a fixed window, say March-May. However, when used as a phenological predictor, this leads to what we call the "time travel problem" in which the phenological event can occur before the fixed window ends, or even begins, so the event's date is predicted by measurements taken after the event. Although we are not using temperatures to directly predict phenological events, we still think it is more appropriate to use more flexible measures of spring temperatures. We use the date on which the 48-day running-average temperature first exceeded 13° C as our measure of spring temperature. The values of 48 days and 13° C give the lowest root-mean-square error (RMSE) between predicted and observed dates on which the Trelease Woods canopy reached 95% closure. Since these values are calculated for the canopy as a whole, we do not have separate calculations for the different understory species. As with the over-time analysis, we regressed this date against the four limiting factors and their sum for each analyzed species.

To bring our light calculations one step closer to the demographic processes that ultimately matter for plants, we calculated carbon-assimilation consequences of phenological shifts for species with suitable published photosynthetic data. Two studies [4, 6] were identified reporting seasonal trends in photosynthetic parameter estimates for understory plants of species represented in our dataset. One of these studies [6] included environmental variables (soil moisture and vapor pressure deficit) in its models that we lacked for our study site, so we did not include the single overlapping species (*Acer saccharum*) from that study. This left four herb species from our dataset with suitable published photosynthetic parameters [4]: *Arisaema*

*triphyllum*, *Asarum canadense*, *Claytonia virginica* and *Hydrophyllum virginianum*. For *H. virginainum*, spring and autumn leaf cohorts were calculated separately. For these species, we computed daily gross total photosynthesis based on modeled understory light values computed from observed above-canopy solar radiation (see Methods Section 1f) and our site-specific canopy light transmission model (see Section Methods 2). These calculations were made using the published photosynthetic parameters from the nearest calendar date. Note that the published source [4] did not distinguish between the two *H. virginianum* cohorts, so we used the closest date of the reported combined data in all cases. We report gross photosynthesis because our initial calculations and published literature [43] both suggest that the roughly monthly intervals of published photosynthetic values [4] do not capture the complex and relatively rapid acclimation processes of respiration to ambient temperature. We evaluated these gross photosynthesis values for long-term trends and trends related to springtime temperatures, similarly to our light calculations.

## Results

For all but one understory species, understory phenology (*U*) and canopy phenology (*C*) were the two most limiting factors for yearly understory light interception, based on their high inter-annual standard deviations (Fig 3). The exception was one herb species active in autumn-winter. Which of these two factors was bigger varied among seasonality groups and species. A consistent pattern arose among all sapling species and all semi-evergreen herb species; for these species and the one winter annual, light interception was more limited by canopy phenology than understory phenology (Fig 3). Likewise, for six of eight spring ephemeral species and six of nine spring-summer species, canopy phenology was the more limiting factor (Fig 3). In contrast, for eight of ten spring-autumn species, the one autumn-winter species, and the one winter perennial species, understory phenology was the more limiting factor for their light interception (Fig 3). For all but two winter-active species, solar radiation (*R*) was the third most limiting factor. Cold temperature (*T*) was the least limiting factor for all species, except for two winter-active species (Fig 3).

Understory and canopy phenology limitations on understory plant light interception very often had opposite signs within a given year; years when this was not true typically had one or both of these factors with small magnitudes (S1 and S2 Figs). Note that in S1 and S2 Figs, a positive value for understory phenology means that understory plants grew their leaves relatively early, thus increasing their potential sunlight interception; a positive value for canopy phenology means late leaf out by trees, increasing light reaching herbs and saplings. Thus, the predominance of opposite impacts of canopy and understory phenology indicates that shifts in the understory and canopy leafy periods were generally in the same direction, i.e., either both strata were early or both were late.

None of the four individual limiting factors separately showed a statistically-significant trend over time for any sapling species (S3 Fig; n.b., S3 and S4 Figs show trends over time and spring temperature, respectively, in all four factors and their combinations individually for each species). However, for blue ash, the overall trend accounting for all factors, and also for only the two phenological factors, was for a significant decrease in light interception over time (S3 Fig). Among the herbs, the separate contribution of the four factors to understory plant light gain was generally non-significant (S4 Fig). Only one species had a significant trend toward more light interception related to understory phenology, none related to cold temperatures, and three species related, all toward less light interception, to canopy phenology (S4 Fig). Interestingly, five species showed a significant increase in light interception over time due to solar radiation (S4 Fig). The phenology-only trend over time from summing canopy

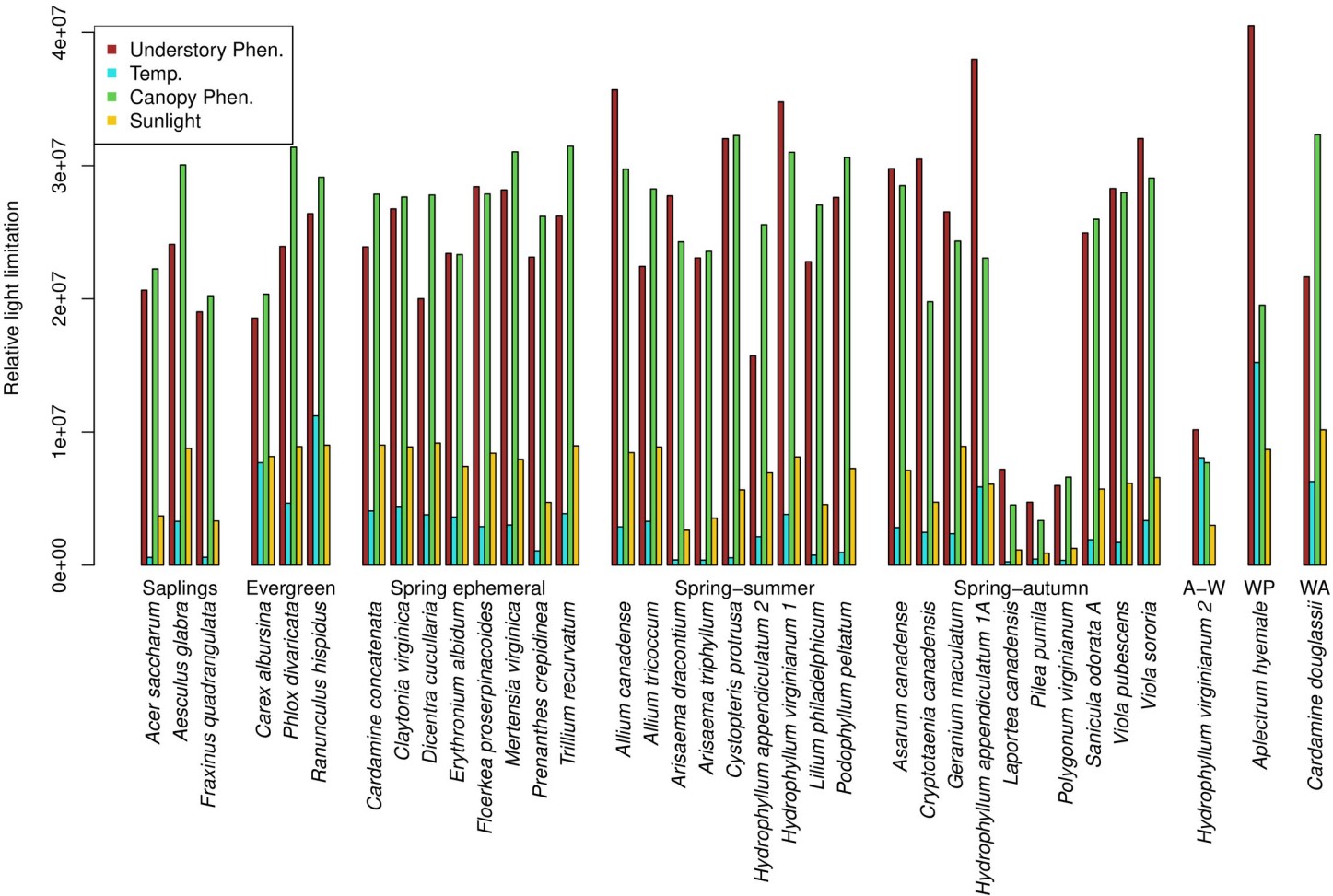

**Fig 3. The relative importance of four factors in limiting the light interception of understory plants.** The bars are the standard deviations of the corresponding bars in S1 and S2 Figs. The species are grouped by life history and seasonality, as indicated between the bars and species' names. Abbreviations: A-W: autumn-winter, WP: winter perennial, WA: winter annual.

and understory phenology (gray symbols and lines in S4 Fig) was significant for 10 species, toward less light interception in nine of these cases. The overall (four factor) trend was significant for six species, five of them toward less light interception. Two species showing a significant overall trend did not show significant trends in any individual factor, and two species showed a significant overall trend in the opposite direction of a significant individual factor. In some cases, multiple individually-insignificant factors summed to a significant trend. The phenology-only trend generally had a steeper slope than the overall trend, indicating a role of sunlight and cold in mitigating the combined impact of canopy and understory phenology.

When analyzed as a function of spring temperature rather than a simple trend over time, significant patterns in the impact of the four limiting factors were much more common (S5 and S6 Figs). Greater light interception due to shifted understory phenology was significantly associated with warmer spring temperature for all three sapling species and 18 of 33 analyzed herb species; in all cases, warmer temperatures led to more light interception (S5 and S6 Figs–note that in these figures warmer years are to the left and cooler years to the right). Cold temperatures were less limiting to light interception in warmer years for one sapling species and 11 herb species. Canopy phenology became significantly more limiting in warmer years for all

three sapling species and 31 of 33 analyzed herb species. Finally, for all three sapling species and 28 herb species, a positive impact of solar radiation on light interception was significantly correlated with warmer years (S5 and S6 Figs). In aggregate, these variables tended to cancel out one another; only one sapling and seven herb species showed overall trends of significantly less light interception with greater spring warmth (S5 and S6 Figs). When only phenological factors (*U* and *C*) were evaluated, warmth-related trends tended to be stronger (note that the gray lines tend to be steeper than the black lines in S5 and S6 Figs).

Of the five herb species-cohort combinations (hereafter "species") with suitable photosynthetic data, three showed statistically significant trends over time in gross $CO_2$ assimilation (S7 Fig). *Arisaema triphyllum* and *Hydrophyllum virginianum* cohort 2 increased their modeled gross C assimilation between 1995 and 2017, while *Asarum canadense* gross C uptake declined over that period (S7 Fig). Interestingly, these three species all showed light-interception trends in the same direction, insignificantly for *Arisaema* and *Hydrophyllum*, and strongly so for *Asarum* (see S4 Fig). When evaluated as a function of integrative spring temperature, no species showed a statistically significant trend in either direction (S8 Fig). Although the weaker patterns for spring temperature compared to over-time changes appear to counter the broad trend shown in light interception calculations, this seems to be a matter of a small sample size of species with available photosynthetic data. All of these species showed similarly weak overall light interception trends related to spring temperature (see S6 Fig).

## Discussion

This study has examined the consequences of phenological shifts on understory plants' light interception and gross photosynthesis, using nearly three decades of observations of marked saplings and trees, and herbs in fixed plots, totaling dozens of species by a single observer in a single temperate deciduous forest. This unique dataset, along with local weather and light observations, allows unprecedented inference on how four factors (understory phenology, cold temperatures, canopy phenology, and solar radiation) affected light interception by over 30 species of plants growing in the understory. As was expected from prior analysis of these phenological data sets [8], the many understory species, with diverse growth forms, phenological patterns, and life histories, show idiosyncratic results and do not fit within a simple, generalized explanation. But, overall, this study's results have shown a trend that, for most species, in years with increasingly warm springs, canopy phenology tends to limit light reaching herbs and saplings at most only slightly more than a majority of understory species' phenological shifts can make up for, similar to Scenario 1 (Table 1). However, this minor asymmetry is further mitigated by increased sunlight and fewer periods in which cold temperatures limit potential photosynthesis in warm years, as in Scenarios 4 and 6 (Table 1). This is generally contrary to the mismatch hypothesis summarized in Scenario 3 (Table 1). However, it is important to remember that light interception alone does not directly determine the demographic success of plants. Rather, it is an intermediate link in a chain through gross carbon assimilation, itself a highly non-linear function of light, to respiration and net carbon gain, survival, growth, flowering, seed production, and ultimately next-generation seedling establishment. The role of respiration in this process should not be underestimated; although warmer temperatures may reduce photosynthetic limitations, they also increase respiration in complex ways [43]. All of these processes can be further upset by changes in pathogens or herbivores. Taken together, we conclude that the overall patterns found in this study show that changes in phenology alone are unlikely to have a big impact on the future of temperate forest understory biodiversity.

Temperature is clearly an important phenological cue for plants, but it interacts with other cues in complex, species-specific and imperfectly understood ways [44]. This complicates predicting how phenological changes, including phenological mismatches, will develop in the future. Air at canopy level is generally colder than air in the understory during the winter and early spring when temperate deciduous forest understories are brightest, and soil temperature changes are lagging and muted responses to air temperature. However, these differences do not mean that species cued by temperatures in relatively cool environments will have inherently smaller responses to global warming. Just as important as absolute changes in relevant temperature environments is per-degree sensitivity of plants to these changes and their inherently non-linear [45, 46] responses to further warming. The general lack of detailed understanding of phenological cue-response patterns, particularly of species- and life-stage-specific responses, means that all possible responses to pervasively warmer temperatures are possible. This study shows which responses are really happening.

Among the four factors potentially limiting understory light availability, canopy and understory phenology were clearly the most influential, modified by the secondary factors of solar radiation and cold temperatures. Whether canopy or understory phenology was a bigger factor was not consistent among seasonality groups and sometimes also inconsistent among species within seasonality groups. Furthermore, the four factors affecting light showed insignificant trends over time for most species, either separately or combined. Overall, there was no consistent trend toward increasingly limited light availability of understory plants in the later years of the study, and minimal trends toward more light limitation in years with warmer springs.

All species with significant trends in their limitation by cold temperatures were less limited in warmer years, although these slopes were never as steep as for the phenological factors. It is intuitive that limitation of light interception by cold temperatures would be reduced in warmer years, but it is not a certain outcome, and there are many reasons we advise against reading too much meaning into this result. Some models have predicted increasing variability in temperatures as the climate warms [47]. In our study area, February and March have recently had high interannual variation that included some extreme weather years [48]. High interannual variability can obscure long-term patterns [47], indicating the need for studies even longer than this one. Variability in spring temperatures can lead to very cold periods after leaves have begun to appear [48], damaging tissues [49], and ultimately reducing light interception and affecting survival [50].

We chose not to report trends over time in solar radiation intensity in the Results because 7.9% of days have missing or questionable values (see Methods Section 1f), and due to our own lack of expertise in this type of data. However, for its relevance to interpreting our other findings, we note that we did not find any significant trends in solar radiation over the period 1995–2022. If anything, a slight dimming might be expected over this period due to a general decrease in sunspot activity [51], but this could be complicated by changing cloud-cover patterns. Indeed, that all significant relationships between spring warmth and solar radiation's contribution to understory light interception were positive suggests that warmer springs may have been less cloudy over the data-collection period. Even if this is not true, solar radiation's hierarchical ranking below the other three limiting factors means sunlight can still show changes in its relative importance depending on how much room the other factors' variations leave for solar radiation to matter, even if solar radiation itself shows no long-term trend. Note that when sunlight did show a significant trend over time as a part of a species' light budget, its slope was substantially shallower than the slopes of understory and canopy phenology (S4 Fig).

Although this study builds on an unprecedented combination of long-term datasets from a single site, certain assumptions were still unavoidable, and may have consequences for interpreting our results. Our assumption that canopy tree leaf area (or at least light interception)

was proportional to basal area in a similar way across species has some published precedent [38], but we cannot be certain how consistent the tree species in this study site are. Given the relative diversity of the canopy (sugar maple, the most common species, accounted for <20% of basal area at the beginning of the study period), we cannot think of a way this assumption could lead to systematic bias in our results. A related assumption is that saplings have similar relationships between phenological stage and light interception ability as canopy trees. If this assumption is imperfect, it should at least be consistently imperfect over time or different spring temperature regimes, so would not be expected to bias our longitudinal analyses. The diversity of the canopy does pose a problem in that understory light environment depends heavily on the species above an understory plant. It makes a big difference for a sapling if the canopy trees above it are mostly early- or late-leafing species. Lacking a detailed stem map, we could not make an even approximate correction for this variable, so our results are indicative of understory-wide averages. Local microsites will obviously vary.

We do not have strong evidence for our assumption that all species have the same minimum cutoff temperature for photosynthesis. Rather, species differ in their inhibition of photosynthesis by cold temperatures, and adjust this inhibition dynamically [41, 42]. If some species are particularly sensitive to chilling at relatively high temperatures, that would not be reflected in our results. However, the generally small effect found for cold temperature limitation means that this would have to be a big difference among species for it to matter much.

Similarly, we are not aware of any research that provides good evidence for or against our assumed linear decline of light interception ability with senescence stage by saplings in autumn. However, as springtime light contributes much more to sapling light budgets than autumn light, this assumption probably does not have a huge impact on our results.

Probably the most consequential of our assumptions is that light interception is a useful proxy for carbon balance and ultimately plant demographics. Respiration is probably the biggest complication in this story. Whole plant compensation points are hard to measure, but, at least at a leaf level, some understory plants can indeed maintain a positive carbon balance even in the deep shade of summer [4, 5]. When we focus on cold temperature limitation of plants' ability to make use of light, we want to emphasize that we are not focusing on all aspects of temperature dependence of carbon balance. Warm temperatures' impacts on respiration are probably more important than low temperature limitations, but detailed physiological research into this issue is well beyond the scope of this study. However, even though photosynthesis is at best non-linearly related to light intensity, for the five herb species with available photosynthetic data, we found a broad similarity between light interception and gross photosynthesis trends, related to both changes over time and to spring temperature. This indicates that the light-related results can be cautiously extrapolated at least to gross carbon gain for other understory species. Even with a detailed understanding of carbon balance, something that we do not pretend to provide here, other factors such as pathogens and herbivores can muddy links to plants' demographic outcomes. Questions surrounding these various processes are all important. We hope to have brought more clarity to a few of them.

The results presented here find at best weak support for the mismatch hypothesis (Scenario 3 in Table 1). Only a small minority of herb species and one sapling species demonstrated the predicted pattern. Earlier canopy formation in warmer years did consistently limit understory plants' light availability (S5 and S6 Figs), but most sapling and herb species mostly (but not entirely) offset this limitation by leafing earlier. The small loss due to mismatched canopy and understory phenology was further compensated by reduced cold limitation and more sunlight (Scenarios 4 and 6 in Table 1). These results demonstrate that understanding the limitation of potential phenological asynchrony or mismatches is more nuanced when secondary environmental factors related to the causes of phenological shifts are also considered.

The few species showing clear patterns of less light interception with warmer springs may show population declines in future years, with potential impacts on biodiversity of the forest, depending on how co-occurring or in-migrating species respond. However, given the substantial impacts from deer herbivory seen in Trelease Woods [37] and many other temperate forests globally [30], we suspect that such patterns will be at most secondary drivers of forest understory biodiversity loss.

The gross carbon assimilation trends were broadly congruent to light interception patterns within the same species. However, the species with available photosynthetic data are somewhat unrepresentative of the most commonly seen patterns of changes in drivers of light interception among all studied herb species. For us, this is reason to be cautiously optimistic that the patterns seen in light interception will indeed translate into similar patterns in understory plant demography, despite the complex non-linear relationship between light interception and photosynthesis. However, it is important to remain aware of the many additional factors that will inevitably complicate demographic outcomes even further. The previously noted complexity of respiration that led us to report gross rather than net photosynthesis is probably the largest [43]. Understanding understory plant photosynthesis would benefit greatly from a better understanding of the environmental drivers of respiration, although this appears to be a challenging task. Additional complexities include how carbon assimilation translates into growth and reproduction, and how these processes can always be short-circuited by pathogens and herbivores. Despite these complications, we believe that this study is an important step toward clarifying how phenological changes brought on by global warming will impact temperate forest understories.

Extrapolating into a future with even greater spring and autumn warmth, but possibly also greater extremes, it is possible that a pattern will arise with some understory species receiving slightly less light in most years, but punctuated by extreme years with cold temperatures after much leaf development of both strata, as occurred in 2007 [49]. However, the warming-related trend of increasingly early phenology in the last century may have slowed in recent decades [52, 53]. In addition, recent autumn/winter warming may delay spring phenology because of inadequate breaking of dormancy [54]. Given the variability of weather patterns [47, 55], climate change has not been linear. Furthermore, the inherent non-linearity of warmth-accumulation processes [45, 46] may mean persistently non-linear relationships between weather and phenology [52, 56]. All of these factors may weaken any association of the temporal trends between canopy and understory phenology and their effects on light limitation of understory plants.

## Supporting information

**S1 Table. Species of canopy trees used in this analysis, the number of living trees of each species in the census in 1995 and 2022, and the relative basal areas of these species in Trelease Woods at two times during the study period.** These basal areas values were normalized so that they sum to 100%, which in our analysis has the effect of treating other tree species (representing <4% of basal area) as though they have phenology that is equivalent to the overall average of the observed species. There was minimal change in species' basal area from 1995–2015. Relative basal areas changed dramatically when the emerald ash borer killed virtually all ash trees in Trelease Woods between 2015–2018. Note that three Ohio buckeye trees were added to the census in 1996, hence the apparent increase from 1995 to 2022. (DOCX)

**S2 Table. Description of events within each phenophase used to quantify light interception.** On a given census date, the stage documented represents the tree crown as a whole, i.e.,

the dominant condition of buds, shoots or leaves.
(DOCX)

**S3 Table. Herb species-cohort combinations (referred to as "species" in the main text) observed from 1995–2017 in Trelease Woods.** Also shown is the phenological seasonality of each species.
(DOCX)

**S4 Table. Relative basal area of trees with a diameter at breast height greater than 22.86 cm (9 inches), by species (rows) for all years of the studies (columns).** This dataset is based on a 2005 census by J. Edgington, and is modified from 2016 onward to account for the death of ash trees because of the emerald ash borer.
(CSV)

**S1 File. Details of the herb data cleaning process.**
(DOCX)

**S2 File. Graphical explanation and full equations for the light-partitioning process applied in this study.**
(DOCX)

**S1 Fig. The relative impact of sapling phenology, temperature, canopy phenology and solar radiation on sapling species' light interception, by year.** The y-axis units are relative measures of light interception, and best used for comparisons within species (see Methods: Section 4).
(PDF)

**S2 Fig. The relative impact of herb phenology, temperature, canopy phenology and solar radiation on herb species' light interception, by year.** The y-axis units are relative measures of light interception, and best used for comparisons within species (see Methods: Section 4).
(PDF)

**S3 Fig. Trends in total light interception by sapling species over time, and relative contribution to that trend of sapling phenology, temperature, canopy phenology and solar radiation.** The y-axis units are relative measures of light interception, and best used for comparisons within species (see Methods: Section 4). Solid lines indicate a factor has a statistically-significant ($p < .05$) difference of its estimated slope from 0, while dashed lines indicate that this standard was not met.
(PDF)

**S4 Fig. Trends in total light interception by herb species over time, and relative contribution to that trend of sapling phenology, temperature, canopy phenology and solar radiation.** The y-axis units are relative measures of light interception, and best used for comparisons within species (see Methods: Section 4). Solid lines indicate a factor has a statistically-significant ($p < .05$) difference of its estimated slope from 0, while dashed lines indicate that this standard was not met.
(PDF)

**S5 Fig. Trends in sapling species' light interception as a function of date on which the 48-day running average temperature first exceeded 13° C (see Methods: Section 6).** This integrative measure of spring temperature means that warmer springs fall to the left on the x-axis. Solid lines indicate a statistically-significant ($p < .05$) difference of the estimated slope

from 0, while dashed lines indicate that this standard was not met.
(PDF)

**S6 Fig. Trends in herb species' light interception as a function of date on which the 48-day running average temperature first exceeded 13˚ C (see Methods: Section 6).** This integrative measure of spring temperature means that warmer springs fall to the left on the x-axis. Solid lines indicate a statistically-significant (p < .05) difference of the estimated slope from 0, while dashed lines indicate that this standard was not met.
(PDF)

**S7 Fig. Trends in gross annual photosynthesis by herb species by year.** The first page shows an example year (2004) of calculated daily gross photosynthesis, with canopy light transmittance indicated in the background in gray. The remaining panels show these daily values summed for each year, and how that value has changed over time. Solid lines indicate a statistically significant (p < .05) difference of its estimated slope from 0, while dashed lines indicate that this standard was not met.
(PDF)

**S8 Fig. Trends in herb species' gross photosynthesis as a function of date on which the 48-day running average temperature first exceeded 13˚ C (see methods: Section 6).** This integrative measure of spring temperature means that warmer springs fall to the left on the x-axis. Solid lines indicate a statistically-significant (p < .05) difference of the estimated slope from 0, while dashed lines indicate that this standard was not met.
(PDF)

## Acknowledgments

David Zaya's earlier extensive separate analyses of these herb and tree phenology data provided the perspective for this study and his critical comments improved earlier versions of the manuscript. Jack Justus gave feedback on how the light partitioning equations were presented. Mason Heberling and Benjamin Lee gave timely and detailed responses to our queries about incorporating data from their studies into our analyses. Three anonymous reviewers provided valuable feedback on the first submitted version of this manuscript.

## Author Contributions

**Conceptualization:** Carol K. Augspurger.

**Formal analysis:** Carl F. Salk.

**Investigation:** Carol K. Augspurger.

**Methodology:** Carl F. Salk.

**Writing – original draft:** Carol K. Augspurger, Carl F. Salk.

**Writing – review & editing:** Carol K. Augspurger, Carl F. Salk.

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
