## [Decision Letter · Decision Letter 0]

22 Nov 2023

PONE-D-23-26311Understory plants reduce light loss in a temperate deciduous forest amid climate variability by shifting phenology in synchrony with canopy treesPLOS ONE

Dear Dr. Salk,

Thank you for submitting your manuscript to PLOS ONE. After careful consideration, we feel that it has merit but does not fully meet PLOS ONE’s publication criteria as it currently stands. Therefore, we invite you to submit a revised version of the manuscript that addresses the points raised during the review process.

We look forward to receiving your revised manuscript.

Kind regards,

Kunwar K. Singh

Academic Editor

PLOS ONE

A copy of your manuscript showing your changes by either highlighting them or using track changes (uploaded as a *supporting information* file

A clean copy of the edited manuscript (uploaded as the new *manuscript* file).

Reviewers' comments:

Reviewer's Responses to Questions

**Comments to the Author**

1. Is the manuscript technically sound, and do the data support the conclusions?

Reviewer #1: Partly

Reviewer #2: Yes

Reviewer #3: Yes

2. Has the statistical analysis been performed appropriately and rigorously? 

Reviewer #1: Yes

Reviewer #2: Yes

Reviewer #3: Yes

3. Have the authors made all data underlying the findings in their manuscript fully available?

Reviewer #1: No

Reviewer #2: Yes

Reviewer #3: Yes

4. Is the manuscript presented in an intelligible fashion and written in standard English?

Reviewer #1: Yes

Reviewer #2: Yes

Reviewer #3: Yes

5. Review Comments to the Author

Reviewer #1: PLOS ONE manuscript review

I reviewed the manuscript titled “Understory plants reduce light loss in a temperate deciduous forest amid climate variability by shifting phenology in synchrony with canopy trees” for the journal PLOS ONE. The authors of this paper investigated four different factors that could limit understory plant access to spring light, a critical limiting factor for the growth, survival, and reproduction of various understory species. The authors found that the majority of understory plant species were able to track light availability with their phenology, suggesting that most species should be capable of maintaining access to this critical resource, with a small minority of species predicted to lose substantial access to spring light under warmer spring conditions.

I want to start off by congratulating the authors on writing a paper that was surprisingly fun to read given how methods-intensive it was. This is the type of research where it really directly translates how I (and I hope others) tend to think about ecology in all its complicatedness in an effort to get meaningful insight from a messy web of information. I was also impressed by the dataset itself, novel in its phylogenetic breadth of species studied, especially considering the nearly 3 decades that data were recorded over. Overall, however, I did have some concerns related to the writing and analysis.

Starting with the writing, I wanted much more discussion than was provided. You have just under 7 pages devoted to results and discussion compared to the more than 21 pages of methods. While I appreciate that the methods section is necessarily long to adequately address all of the assumptions and statistical nuances employed in your analysis (a benefit to the paper, in my opinion), I would like to see more discussion on how your many assumptions could have affected your overall findings. This is particularly important because I felt some of your assumptions were overly simplistic (see analysis comments starting in next paragraph), and could definitely have skewed your results in various ways that should be addressed. On a less important note, I point out in my line comments several places where you seem to have overlooked various papers that do in fact address some of the knowledge gaps presented in the introduction.

In terms of analysis, I want to first say that I appreciate the overall approach that you take in this paper. It seems to me that you started with a list of potential limitations to understory light availability and then pieced together an extensive analysis that would allow you to address all of them within the same framework – very ambitious and, in my opinion, a very worthwhile attempt at it. And while the amount of information was a strength of this paper, I found that the number and quality of assumptions needed to leverage this information was simultaneously a weakness.

Throughout the paper, you make clear that you are only addressing the light interception by understory plant species, stopping short of extrapolating out to carbon gain, plant performance, etc. You even go so far as to state on L551 that “We recommend interpreting differences among species in the relative sizes of the limiting factors cautiously. They can be used to indicate the variability, and thus relative importance of the different factors to individual species, but, due to species’ physiological differences, should not be seen as direct indicators of downstream consequences such as carbon sequestration.” While I think you are correct in this statement, I also think that this statement takes a lot of importance away from this research. The reason why light availability is important is because it directly affects carbon sequestration, so there is really no way to address this topic without coming back to carbon in the end. For me, I wanted to see more in the discussion extrapolating out to carbon dynamics and I wanted to see more in the methods addressing how some of your simplifying assumptions are not realistic in terms of how plants in this system assimilate carbon.

One example of this is that many understory plant species (particularly those that maintain their leaves into summer) adjust their photosynthetic machinery in ways that allow them to maintain positive carbon gain despite extremely low light levels. In fact, many understory plant species (woody and herbaceous both) have been shown to maintain positive carbon gain throughout much of summer despite full canopy shading (Heberling et al., 2019; Lee & Ibáñez, 2021b, 2021a), indicating that that there is no neat cutoff at which point light becomes limiting. Furthermore, even if there were such a cutoff, it would differ substantially among species – light compensation points (the light level at which carbon lost to respiration equals the carbon gained from photosynthesis) differ drastically among species and are considered a major component of shade tolerance characteristics for understory plant species. Thus, even if you do not want to express overall results in terms of carbon budgets, you should still account for these dynamics in the assumptions of your model.

A really good example of why this important is from one of the papers you missed in your intro. Lee & Ibáñez (2021a) found that warmer springs would lead to greater understory light access for temperate deciduous tree seedlings, which on its face would suggest greater carbon gain in future decades. However, after accounting for photosynthetic rates, and especially after accounting for how respiration responds much more strongly to increases in temperature compared to photosynthetic carbon assimilation, those authors found that seedlings would lose proportionally more carbon from summer respiration than from spring carbon gain. Thus, light availability only communicates part of the overall picture and can be largely meaningless without the photosynthetic information to support it.

I mention all this because of your emphasis on potentially mitigating effects of warmer temperatures. Warmer temperatures can just as easily exacerbate negative effects of shading or reduce the positive effects of increased light availability. I did not feel like your characterization of temperature as mitigating painted a full picture, especially since you invoke photosynthetic limitations without including photosynthesis in your analysis.

If you wanted to, photosynthetic information is published and available for several of the species you use in this analysis. For example, Heberling et al. (2019) addresses photosynthetic carbon gain and how it is affected by access to spring light in several understory herbaceous species. Even if this information is not available for each of the 40-odd species in your analysis, I think your analysis would benefit from picking even just one or two species as case studies to show that light availability is meaningful. In other words, you address four limitations related to light interception, but the fifth, implied assumption is that plants are using that intercepted light for positive carbon assimilation, which is not necessarily the case. To paint a complete picture of how light availability changes and of why that is important, I think you need to at least partially address photosynthesis and carbon assimilation.

Another example, not to beat a dead horse, is that many plants respire much more than they photosynthesize during leaf expansion, meaning that having leaves is not enough to be considered photosynthetically useful. However, the duration of leaf expansion is changing with climate change on a species by species basis (Buonaiuto & Wolkovich, 2021), meaning that some species, if their duration of leaf expansion is growing, may actually be losing days of activity under high light availability, even if the amount of light they are exposed to is increasing. Again, the fifth limitation that you are missing in this analysis is photosynthetic activity. You can’t go back in time and collect this data, but you can (at best) try to quantitatively address this by using others’ data collected on the same species or (at worst) address the importance of these assumptions and how they affect your interpretation of the results in discussion.

I’ll finish by acknowledging that I just did the thing that I hate when reviewers do to me, which is to make the entire review about something that is beyond or other than the explicitly stated goal of the paper. I get it, and I apologize (to a certain extent) for doing so. However, I really do think that you cannot tell the full story on overstory/understory phenological mismatch without addressing photosynthetic carbon assimilation. Even if your main conclusions (related to changes in understory plant light exposure) stand on their own, which I think they do for the most part, I think it is important and necessary that you fully address the potential implications for plant performance mediated through carbon assimilation. I have added line comments below.

Line comments

L58-66: Authors could add references for similar dynamics in juvenile tree seedlings (Lee & Ibáñez, 2021a, 2021b). This would be especially helpful in linking spring light access to growth and survival of woody plants.

L94-95: See (Lee & Ibáñez, 2021b, 2021a) for papers that address entire growing season dynamics for two species of tree seedlings.

L114-118: Again, see papers by Lee and Ibanez for examples of comparing understory phenology directly to observed light levels.

L127-131: Again, see papers by Lee and Ibanez for existing examples of what you cite as not yet existing (regional studies; (at least partially) accounting for differences in canopy composition; etc.)

Paragraph beginning L166: All data should include units of measurement, especially since Fig. 1 is pretty low resolution and even zooming in does not allow for the reader to easily interpret the units shown on the axes.

L170: “Government agencies” could be a bit more specific, even if it is listed later in the methods as well.

Section 2b (beginning L364): It is unclear to me, as currently written, at which spatial resolution basal area was aggregated. Did you include all trees in the 50m x 50m plot? Or did you account for trees that were within a certain distance of the point measurements only? Given the difference in average leaf out times among canopy tree species (e.g., cherry and maple are much earlier than oaks), it makes sense to me to try to pick a distance around a quadrat and include only the trees with basal area overlapping or within that set distance. Is this possible? If not, can you explain more in the text about how the assumptions you make here could affect the analysis?

L523-527: You focus here on cold-limitations to photosynthesis, but temperature effects on photosynthesis are much more complicated. For example, positive carbon gain is impossible at high temperatures because respiration costs scale more steeply with increasing temperatures than does carbon assimilation. Temperature also interacts with light availability for similar reasons – darker conditions require lower temperatures in order for positive carbon assimilation (which only occurs with high-enough light availability) to outweigh negative respiration costs (which are an issue all day long, including overnight when light is never available). Freezing limitations are good to include, but you are potentially under-counting the capacity for photosynthesis to be limited by other aspects of temperature, which I think could potentially lead to an underestimate in how important temperature is as an overall limitation.

L528-36: Species differ significantly in how much light is needed to conduct positive photosynthesis (often referred to as the light compensation point). In this sense, a canopy transmittance value of ~0.1 (in your analytical framework) may be entirely limiting for one species, but not at all limiting for another, even under conditions that are otherwise exactly the same. This is, importantly, a large component of what is considered when researchers quantify shade tolerance among different species. Furthermore, it often changes throughout the growing season in species whose leaves are maintained for longer than a few months. Species-level photosynthetic efficiency information is missing from this estimate of canopy light transmittance limitations and could lead to biased estimates of how much of a limitation this variable causes. Specifically, this limitation would be over-emphasized for shade-tolerant species (which can maintain photosynthetic efficiency even in full shade) and under-emphasized for shade-intolerant species for which even a small decrease in understory light availability could result in complete loss of positive carbon gain. I understand that your goal for this paper is not to extrapolate all the way to carbon budgets, but this is a major (and in my experience, incorrect) assumption that all of these species are equally light-limited. If possible (especially if available from elsewhere in the literature), species-specific photosynthetic efficiencies should be incorporated to account for light limitations. If that is not possible, the implications from this assumption should be clarified in the text.

L551-555: You write “We recommend interpreting differences among species in the relative sizes of the limiting factors cautiously. They can be used to indicate the variability, and thus relative importance of the different factors to individual species, but, due to species’ physiological differences, should not be seen as direct indicators of downstream consequences such as carbon sequestration.” This seems to contradict some of your earlier justification for why this research is novel and important. Photosynthetic parameters are likely available for some of these species from other manuscripts (see, for example, Heberling et al., 2019). Would it be possible to extrapolate these “downstream consequences” for at least a few select species where this information is available? For me, it would go a long way in justifying the rest of this very complicated approach.

L696-697: This holds true only in winter, whereas the opposite is generally true in summer.

L728-730: Isn’t it more likely that this positive association comes from higher temperatures on non-cloudy days? Meaning that past springs with fewer clouds are likely to be warmer, but not necessarily that warmer spring temperatures will reduce future cloudiness.

L741-742: This compensation may not be valid considering you are not accounting for high temperature limitations to photosynthetic efficiency (see Lee & Ibáñez, 2021a for an example of how high temps are exacerbating, not ameliorating to overall carbon dynamics in warmer summers).

Papers Cited:

Buonaiuto, D. M., & Wolkovich, E. M. (2021). Differences between flower and leaf phenological responses to environmental variation drive shifts in spring phenological sequences of temperate woody plants. Journal of Ecology, 109(8), 2922–2933. https://doi.org/10.1111/1365-2745.13708

Heberling, J. M., Cassidy, S. T., Fridley, J. D., & Kalisz, S. (2019). Carbon gain phenologies of spring-flowering perennials in a deciduous forest indicate a novel niche for a widespread invader. New Phytologist, 221(2), 778–788. https://doi.org/10.1111/nph.15404

Lee, B. R., & Ibáñez, I. (2021a). Improved phenological escape can help temperate tree seedlings maintain demographic performance under climate change conditions. Global Change Biology, gcb.15678. https://doi.org/10.1111/gcb.15678

Lee, B. R., & Ibáñez, I. (2021b). Spring phenological escape is critical for the survival of temperate tree seedlings. Functional Ecology, 1365-2435.13821. https://doi.org/10.1111/1365-2435.13821

Reviewer #2: The study examines whether phenological shifts in 31 herbs respond to four factors that could limit their light interception. This adds to the growing literature investigating how canopy phenology might influence understory herbs. I really appreciated the time spent on why a local study addresses key questions about understory-canopy mismatch that a broad synthesis cannot, as well as the step by step modelling decisions. However, the methods section is extremely difficult to read. I include specific questions below, but overall recommend some simplifying/summarizing statements at the beginning/end of each sub-section (or if the bulk of the text could be simplified and the details moved to an appendix). I could also use some discussion on the collinearity of a lot of these measurements, which are subsequently multiplied together, as well as clearer figures (improved Figure 1 and some presentation of the differences in slopes).

Abstract: It’s a little difficult to follow negative/positive shifts, “beneficial” shifts (especially given there’s no fitness measurement), and synchronous shifts. It would be worth setting up, even in the Abstract, which direction phenology needs to be shifting in order to be synchronized with light, and then use that term to describe the results.

Introduction:

L57: Add citation.

L77: Air temperatures for herbs as well?

L137: While I understand that this last paragraph of the Introduction is further outlining how this study differs from previous work, it feels very much like a methods section. Suggest rewriting to focus on hypotheses – what directions of sensitivities do we expect to see? What should we see if the slopes are mismatched or not? The methods details here should be integrated elsewhere.

Methods

L230: Can you explain this cohort grouping more clearly? What does it mean biologically? Did it make a difference when analyzing species/cohort vs. species?

L277: What was the date choice for light interception here? Was this the same value for all species in the study? Or were different Julian calendar dates for temp limitation used for each species based on some mean leaf-out time? Or is this that every Julian day of the weekly observation had this 4-step process applied?

L349: Is the caption for Fig. 2 supposed to be floating here?

L374: How was tree phenological stage converted to transmittance value?

L395: Optimal values for what? Interception or transmission?

Section 2b is a little hard to follow. Perhaps some summary at the beginning about estimating transmission, then interception, then something about adjusting for shading values?

L407: What does this variance by stage-specific shading do, relative to the equal spacing in the paragraph prior?

L456: Isn’t this leaf area measurement completely colinear with phenology given how it was estimated? This needs more justification.

L461: But this measure of interception was not measured in 2b? Why? What’s the difference?

L523: Why not use continuous temperature from the weather station? Temperature is likely still affecting phenology so this grouping doesn’t make much sense.

L598: Why is more variable == more limiting? Why not a mean low value?

L617: I agree with this logic, but does the date chosen vary for each species?

L619: chosen how?

Results

L634: How would temperature itself be “limiting”, especially if it’s measured as a grouping of 0-5 or >5 (L523)? Would this be better calculated as growing degree days to get at heat limitation?

L648-664: Please rewrite in terms of what’s happening. Negative trend, when it’s unclear whether we’re thinking of the slopes as advances in phenology or decreased light interception, is not informative because the predictor variables are muddled. The last paragraph of the Results (L665) is the most clear.

Discussion

L696: How is this conclusion about temperature drawn?

Figure 1: Is chaotic. Perhaps even just straight arrows would help. “Leaf status” not used in main text.

I was expecting some figure showing differences/mismatches in slopes based on the abstract, or a figure in relation to spring temperatures (L665).

Reviewer #3: Dear Editor,

It was my pleasure to review the manuscript titled, “Understory plants reduce light loss in a temperate deciduous forest amid climate variability by shifting phenology in synchrony with canopy trees” by Carol Augspurger and Carl Salk. In this manuscript, the authors integrate several long-term data sets to estimate how light availability for understory plants has changed over time and to weigh the relative impacts of understory plant phenology, canopy phenology, temperature, and solar radiation on this factor. Using their light availability metric as lens to investigate the potential impacts of warming climate and shifting phenology, the authors find that generally little change has occurred in the amount of light available to understory plants, though some species exhibited idiosyncratic responses.

In my opinion, the results detailed in this paper are a valuable addition to the literature on climate change impacts on plants. The authors make excellent use of unique, long-term, detailed observational data to track light availability over the span of decades. This manuscript is well-written, and the descriptions of data sources and analytical methods are detailed and clear.

I do have some suggestions for how this manuscript might be improved:

1. Figure 1 is difficult to take in. The arrows end up making a tangle that is hard to make sense of, and the axis labels on plots are small and difficult to read. I suggest arrows with right angles might be easier to track. Vertically aligning the plots in the left side “Models” split would also aid readability. The caption describing this figure could also be expanded to add clarity. Walking through panel by panel would be beneficial, I think.

2. I think reader’s understanding of the paper’s results might be improved with a figure or table that walks through expected patterns in parameters under several possible scenarios. The authors should consider including a table with columns for: “Changes in canopy spring phenology”, “Changes in understory spring phenology”, “Changes in canopy autumn phenology”, “Changes in understory autumn phenology”, and for changes in light availability and each of the potential drivers of light availability under these scenarios. With each row representing a different scenario, the authors could indicate whether how variables would change given particular set of phenological shifts. Giving readers a tool such as this to think through how the influence of different drivers might be altered under different phenological scenarios could aid in their interpretation of results. Including such a figure/table might also help readers to better understand how solar radiation’s influence might vary from year to year, as key phenological windows hit at different seasonal timings.

3. I would like to see the plots showing the light availability over the years for each species in addition to the plot showing the breakdown of the potential drivers’ influence over time. These could be included as new supplement. Showing this output would allow readers to clearly see whether there is a trend in light availability over time for the understory species considered here (regardless of which drivers might be influencing that availability).

4. Citations justifying the use of the same temperature thresholds for photosynthesis for all species included in the study should be provided in the paragraph on lines 273 through 289.

5. I am curious if there are other, established practices for partitioning the influence of component variables (U, C, T, R here) to a derived variable (L). While the authors’ method for assessing the importance of sub-factors is described thoroughly, I lack the expertise to evaluate whether their method is entirely appropriate. Citations pointing to other works using similar approaches would be reassuring. Or, alternatively, a brief discussion justifying the need for an apparently novel approach that indicates the shortcomings of existing methods would also help. Because the paper’s conclusions hinge on these values, additional justification for the chosen approach would strengthen readers’ trust in the stated results.

I also have a small number of minor edits to suggest:

L23: I suggest “solar radiation” in place of “sunlight”

L69: I suggest “typically” be inserted before “resulting” to clarify that this is not universally true

L73: Some folks consider different hypotheses, including trophic mismatches, and call them “phenological mismatch hypotheses”. Consider changing or clarifying terminology here.

L138: I suggest “Data sources were available…” to maintain past tense.

L185: I suggest adding a comma after “level”.

L178: I suggest R in place of S for consistency.

L202: What magnification were the binoculars?

L433: I suggest “vary” rather than “drift freely”

L606: I am unclear what “limitation values” are being referred to here. Please clarify what these are.

L633: Do you mean “solar radiation (R)” here?

L703: I suggest softening this statement to, “This study shows which responses are really happening”

6. PLOS authors have the option to publish the peer review history of their article (what does this mean?). If published, this will include your full peer review and any attached files.

Reviewer #1: No

Reviewer #2: No

Reviewer #3: No

---

## [Author Response · Author response to Decision Letter 0]

30 Apr 2024

Please see the uploaded response to reviewers file.

---

## [Decision Letter · Decision Letter 1]

2 Jun 2024

PONE-D-23-26311R1Understory plants evade shading in a temperate deciduous forest amid climate variability by shifting phenology in synchrony with canopy treesPLOS ONE

Dear Dr. Salk,

Thank you for submitting the revised manuscript titled "Understory plants evade shading in a temperate deciduous forest amid climate variability by shifting phenology in synchrony with canopy trees" to PLOS ONE.

The reviewers have provided their feedback, noting that the quality of the manuscript has improved substantially. However, a minor revision is necessary to address the reviewers’ comments fully and meet PLOS ONE publication criteria. You can find the reviewers' comments at the bottom of this letter.

We invite you to submit a revised version of the manuscript.

We would appreciate receiving your revised manuscript by Jul 17 2024 11:59PM. When you are ready to submit your revision, please log on to PLOS ONE Editorial Manager and select the 'Submissions Needing Revision' folder to locate your manuscript file.

Best regards,

Kunwar K. Singh

Please include the following items when submitting your revised manuscript:A rebuttal letter that responds to each point raised by the academic editor and reviewer(s). You should upload this letter as a separate file labeled 'Response to Reviewers'.A marked-up copy of your manuscript that highlights changes made to the original version. You should upload this as a separate file labeled 'Revised Manuscript with Track Changes'.An unmarked version of your revised paper without tracked changes. You should upload this as a separate file labeled 'Manuscript'.If applicable, we recommend that you deposit your laboratory protocols in protocols.io to enhance the reproducibility of your results. Protocols.io assigns your protocol its own identifier (DOI) so that it can be cited independently in the future. For instructions see: https://journals.plos.org/plosone/s/submission-guidelines#loc-laboratory-protocols. Additionally, PLOS ONE offers an option for publishing peer-reviewed Lab Protocol articles, which describe protocols hosted on protocols.io. Read more information on sharing protocols at https://plos.org/protocols?utm_medium=editorial-email&utm_source=authorletters&utm_campaign=protocols.

We look forward to receiving your revised manuscript.

Kind regards,

Kunwar K. Singh

Academic Editor

PLOS ONE

Journal Requirements:

Additional Editor Comments:

Dear Carl,

Thank you for submitting the revised manuscript titled "Understory plants evade shading in a temperate deciduous forest amid climate variability by shifting phenology in synchrony with canopy trees" to PLOS ONE.

The reviewers have provided their feedback, noting that the quality of the manuscript has improved substantially. However, a minor revision is necessary to address the reviewers’ comments fully and meet PLOS ONE publication criteria. You can find the reviewers' comments at the bottom of this letter.

We invite you to submit a revised version of the manuscript.

We would appreciate receiving your revised manuscript by June 15, 2024. When you are ready to submit your revision, please log on to PLOS ONE Editorial Manager and select the 'Submissions Needing Revision' folder to locate your manuscript file.

Best regards,

Kunwar K. Singh

Reviewers' comments:

Reviewer's Responses to Questions

**Comments to the Author**

1. If the authors have adequately addressed your comments raised in a previous round of review and you feel that this manuscript is now acceptable for publication, you may indicate that here to bypass the “Comments to the Author” section, enter your conflict of interest statement in the “Confidential to Editor” section, and submit your "Accept" recommendation.

Reviewer #1: All comments have been addressed

Reviewer #3: All comments have been addressed

2. Is the manuscript technically sound, and do the data support the conclusions?

Reviewer #1: Yes

Reviewer #3: Yes

3. Has the statistical analysis been performed appropriately and rigorously? 

Reviewer #1: Yes

Reviewer #3: Yes

4. Have the authors made all data underlying the findings in their manuscript fully available?

Reviewer #1: Yes

Reviewer #3: Yes

5. Is the manuscript presented in an intelligible fashion and written in standard English?

Reviewer #1: Yes

Reviewer #3: Yes

6. Review Comments to the Author

Reviewer #1: Thank you for your thoughtful responses to my initial comments. The paper looks great and I have no further comments.

Reviewer #3: Dear editor,

Thank you for the opportunity to review the revised manuscript, “Understory plants evade shading in a temperate deciduous forest amid climate variability by shifting phenology in synchrony with canopy trees.” The authors have responded adequately to the suggestions I made in my prior review. Figure 1 has been much improved, and I find the table summarizing some likely scenarios to be a helpful tool for thinking through the interactions between the evaluated drivers of light availability.

I did notice a few small line items the might be improved during my review:

L 135: I think you mean “species-specific” here

L 136: I’m not sure “futuristic” is the right word choice. I suggest, “forward-looking” instead.

L 147: This sentence gets confusing. Consider listing the four factors in parentheses to clarify?

L 167: I think you mean to say the that table does NOT include all possibilities, correct? If so, this needs to be corrected here.

L 669: I suggest “left” in place of “leaves” to maintain past tense

L 827: Consider “cannot” in place of “can’t”

7. PLOS authors have the option to publish the peer review history of their article (what does this mean?). If published, this will include your full peer review and any attached files.

Reviewer #1: No

Reviewer #3: No

---

## [Editor Report · Decision Letter 2]

10 Jun 2024

Understory plants evade shading in a temperate deciduous forest amid climate variability by shifting phenology in synchrony with canopy trees

PONE-D-23-26311R2

Dear Dr. Salk,

We’re pleased to inform you that your manuscript has been judged scientifically suitable for publication and will be formally accepted for publication once it meets all outstanding technical requirements.

Kind regards,

Kunwar K. Singh

Academic Editor

PLOS ONE
---

## [Editor Report · Acceptance letter]

18 Jun 2024

PONE-D-23-26311R2 

PLOS ONE

Dear Dr. Salk, 

I'm pleased to inform you that your manuscript has been deemed suitable for publication in PLOS ONE. Congratulations! Your manuscript is now being handed over to our production team.

Kind regards, 

on behalf of

Dr. Kunwar K. Singh 

Academic Editor

PLOS ONE